# Identification and characterisation of an elusive bacterial enzyme system for chloromethane dehalogenation

Jasmin Bernhardt [1,2,3], Lukas K. R. Hofmann [1,2,3,4], Paul Klemm [5], Nicole Paczia [4], Olivier N. Lemaire [6,7], Stéphane Vuilleumier [8], Tristan Wagner[6,7] & Julia M. Kurth [1,2,3,5] ✉

Chloromethane, a toxic gas primarily produced naturally, contributes to stratospheric ozone destruction. The anaerobic acetogen *Acetobacterium dehalogenans* can utilise chloromethane as a carbon and energy source, but the associated dehalogenase/methyltransferase has remained elusive. Through comparative transcriptomics we identify a gene cluster, *cdmBCA*, which encodes a corrinoid-dependent methyltransferase system distinct from the characterised Cmu system used for chloromethane degradation in aerobic methylotrophs. Biochemical characterisation reveals that the Cdm system reacts with other haloalkanes, but not with methoxylated aromatics, unlike closely related *O*-demethylases. X-ray structural analysis of the protein CdmB shows a hydrophobic channelling system directing haloalkanes towards cobalamin-dependent activation. Homologous proteins are found in anaerobic prokaryotes, particularly within the phyla Bacillota and Asgardarchaeota, suggesting previously unidentified microbial transformation of chloromethane in the environment. Discovery of the Cdm dehalogenation/methyltransferase system sheds light on the microbial contribution to the global chloromethane cycle.

Halogenated compounds, particularly chloromethane (CM; $CH_3Cl$) and other haloalkanes, play a significant role in ozone depletion and climate change[1,2]. CM is a greenhouse gas present in the Earth's atmosphere at an average concentration of approximately 550 parts per trillion[3]. Estimated global emissions range from 4 to 5 Tg[1]. CM significantly contributes to chlorine-dependent ozone depletion in the stratosphere[4]. The origins of CM are diverse, with mainly natural but also anthropogenic sources. CM emissions have been documented in various ecosystems, including oceans[5], peatlands[6] and salt marshes[7]. CM is produced by plants[8,9], algae[10], and fungi[11], as well as abiotically through the reaction of chloride with pectin[12]. Anthropogenic activities such as coal combustion, feedstock, or biomass burning also release CM[13]. Other haloalkanes, such as dichloromethane (DCM), iodomethane, and bromomethane, have both natural and anthropogenic sources as well[14]. CM and DCM can be highly toxic and pose significant health risks to humans and animals, targeting the central nervous system, liver, and kidneys[4,15]. Various methods can be employed to remove these contaminants from the environment, particularly from water and soil, including physical (air stripping, adsorption), biological (microbial degradation), and chemical (oxidation, electrocatalysis)

[1]Institute for Molecular Microbiology and Biotechnology, University of Münster, Münster, Germany. [2]Microcosm Earth Center, University of Marburg and Max Planck Institute for Terrestrial Microbiology, Marburg, Germany. [3]Microbial Physiology Lab, Department of Chemistry, University of Marburg, Marburg, Germany. [4]Max Planck Institute for Terrestrial Microbiology, Marburg, Germany. [5]Center for Synthetic Microbiology (SYNMIKRO), Marburg, Germany. [6]Institut de Biologie Structurale, CEA, CNRS, Université Grenoble Alpes, Grenoble, France. [7]Max Planck Institute for Marine Microbiology, Bremen, Germany. [8]Génétique Moléculaire, Génomique, Microbiologie, UMR 7156 CNRS, Université de Strasbourg, Strasbourg, France. ✉e-mail: Julia.kurth@uni-muenster.de

techniques, which are often used in combination. However, further research is necessary to optimise and improve the efficiency of these removal strategies, as well as to develop more effective methods for mitigating the environmental impact of these pollutants[16,17]. Understanding how bacteria degrade these compounds and the role of dehalogenases in this process is thus of major interest[2,18].

Microbial transformation of CM has been described for several bacterial genera mainly associated with an aerobic lifestyle, including *Hyphomicrobium*, *Methylobacterium*, *Aminobacter*, *Leisingera*, *Roseovarius*, and *Pseudomonas*[19,20]. The anaerobic acetogenic bacterium *Acetobacterium malicum* subsp. *dehalogenans* (*A. dehalogenans*) was the first strictly anaerobic bacterium shown to utilise CM as a carbon and energy source for growth[21,22]. More recently, anaerobic bacteria were found to degrade CM and DCM during co-cultivation[23].

Despite the environmental importance of haloalkanes and the ability of many bacteria to dehalogenate these environmental pollutants[20,24–26], the underlying enzymatic mechanisms often remain poorly understood. To date, the CM utilisation system (Cmu) used by aerobic facultative methylotrophic bacteria for growth on CM is the best studied[19].

In this study, we identify a CM-dehalogenating methyltransferase system, termed Cdm, using comparative transcriptomics and enzymatic assays with recombinant proteins heterologously produced in *Escherichia coli*. Similar to other corrinoid-dependent methyltransferase systems[27–32], the Cdm system consists of a substrate-specific methyltransferase (MTI, CdmB), a corrinoid protein (CdmC) and a second methyltransferase (MTII, CdmA). The X-ray structure of the MTI CdmB features an extensive hydrophobic channel network, likely funnelling small haloalkanes to a central cavity, where the dehalogenation reaction catalysed by the corrinoid protein CdmC may occur.

## Results

### Enhanced expression of genes encoding a corrinoid-dependent methyltransferase system (Cdm) in *A. dehalogenans* upon exposure to chloromethane

The acetogenic bacterium *A. dehalogenans* grows by converting various organic compounds, including methoxylated aromatic compounds (MACs)[21,28,31] and chloromethane[21], to acetate. Growth experiments with the MAC syringate, CM, and combinations thereof (Supplementary Fig. 1) showed that *A. dehalogenans* can simultaneously metabolise both substrates, indicating no strong preference for one methyl source over the other and the absence of substrate-dependent inhibition of dehalogenase gene expression. This versatility may confer an adaptive advantage to *A. dehalogenans* by expanding its ecological niche. To identify the enzyme system that enables *A. dehalogenans* to grow on CM, we performed comparative transcriptomics using syringate, CM or both compounds as growth substrates. Comparing the growth of *A. dehalogenans* with CM versus with syringate, 311 genes were upregulated and 271 downregulated ($\log_2$ fold change >2, adjusted $p$-value (padj) <0.05). In contrast, fewer genes were differentially expressed (48 up, 38 down) when CM was compared to the combination of CM and syringate (Supplementary Fig. 2). The combination of syringate and CM versus syringate alone revealed a larger difference in transcription, with 503 upregulated and 176 downregulated genes.

The transcriptional profile of *A. dehalogenans* grown on CM showed an upregulation of genes involved in chloride transport, stress response, and toxin-antitoxin systems, suggesting adaptation to CM-induced stress (Fig. 1; Supplementary Data 1). A growth-inhibitory effect was particularly pronounced for *A. dehalogenans* cultures grown on elevated concentrations of CM and syringate (Supplementary Fig. 1D). Conversely, genes involved in energy metabolism and anabolism were downregulated. This response is reminiscent of that observed in *Methylobacterium extorquens*[33] and *Nitrosomonas europaea*[34], where CM exposure triggers a general repression of metabolism and cell growth, potentially mediated by toxin-antitoxin systems. With regard to the conversion of methyl groups, *A. dehalogenans* encodes 26 putative methyltransferase systems, likely associated with the utilisation of diverse methylated compounds. Comparative analysis of gene expression on CM versus syringate highlighted differential regulation of specific methyltransferase systems (Supplementary Fig. 3, Supplementary Data 1). Some quaternary amine methyltransferase systems are upregulated during CM growth, suggesting a common regulatory mechanism for CM- and quaternary amine-specific methyltransferase systems. Notably, a corrinoid-dependent methyltransferase system encoded by the genes ACIUZZ_RS16855, ACIUZZ_RS16850, and ACIUZZ_RS16845 was strongly upregulated during growth on CM and CM plus syringate compared to syringate alone (Fig. 1, Supplementary Figs. 2 and 3, Supplementary Data 1). These genes encode an MTI, a corrinoid protein (CP), and an MTII, representing a previously unidentified and uncharacterised methyltransferase system, termed Cdm, that was hypothesised to be responsible for CM degradation in *A. dehalogenans*.

### The CdmBC proteins exhibit high activity with chloromethane

We heterologously expressed and purified CdmB and CdmC, whose genes are highly upregulated in response to CM, to investigate their role in CM degradation (Supplementary Fig. 4). Typically, methyltransferase systems also require an ATP-dependent corrinoid activating enzyme (AE) to reduce cobalamin from the Co(II) to the Co(I) state. *A. dehalogenans* encodes a previously characterised AE[28] (ACIUZZ_RS15945; WP_026395886) and three additional putative AEs (ACIUZZ_RS12170, ACIUZZ_RS15240, ACIUZZ_RS16525), none of which was significantly upregulated in response to CM. We hypothesised that ACIUZZ_RS15945, encoding the AE in closest proximity to the *cdmBCA* genes, which had already been shown to activate the CPs from the two characterised *O*-demethylation systems of *A. dehalogenans*[28], might serve as the AE for the Cdm system. UV-vis spectroscopy using CdmBC and AE proteins from *A. dehalogenans* heterologously produced in *E. coli* revealed that AE-mediated reduction of Co(II)-CdmC to Co(I)-CdmC is followed by an MTI CdmB-mediated reaction with CM to form $CH_3$-Co(III)-CdmC (Fig. 2). The observed methyl transfer is consistent with other bacterial methyltransferase systems[29–32]. As we did not succeed in producing CdmA in *E. coli*, the CdmA-mediated tetrahydrofolate ($H_4F$) methylation could not be verified directly in this study. However, previous reports on the dehalogenation of CM by *A. dehalogenans*, based on activity assays using cell extracts[22], demonstrated that the methyl group of CM is transferred to $H_4F$. The MTII that catalyses $H_4F$ methylation is most likely CdmA, as the expected binding site conserves all molecular determinants of $H_4F$ specificity[35] (see AlphaFold 3 structure of the modelled CdmA active site in Supplementary Fig. 5).

In vitro activity assays with MTI CdmB, CP CdmC, and AE (Fig. 2B) revealed that the Cdm system exhibits a high specific activity of Co(III)-CdmC formation with CM as the substrate, only in the presence of CdmB (Table 1). Neither EDTA nor zinc altered this activity, indicating that zinc is not required for catalysis. In contrast, $CH_3$-Co(III) formation from CM and free Co(I)-cobalamin instead of Co(I)-CdmC occurred irrespectively of CdmB, with reaction rates of approximately 64.4 $M^{-1}s^{-1}$ without CdmB and 62.2 $M^{-1}s^{-1}$ with CdmB ($n = 2$ technical replicates), around four times lower than those for the reaction of CM with CdmB-associated CdmC (288.1 $M^{-1}s^{-1} \pm 35.7$, corresponding to a specific activity of $3.9 \pm 0.5\ \mu mol\cdot min^{-1}\cdot mg^{-1}$ CdmB, Table 1). Notably, iodomethane, iodoethane, and bromoethane also reacted with CdmC in the absence of CdmB, likely due to their high reactivity with cobalt. Iodomethane has indeed been shown to be a non-physiological methyl donor for many corrinoid proteins, as it undergoes rapid nucleophilic attack by the cobalt ion[36,37]. Haloalkane reactivity with the Cdm system followed a logical pattern, with iodinated compounds being most

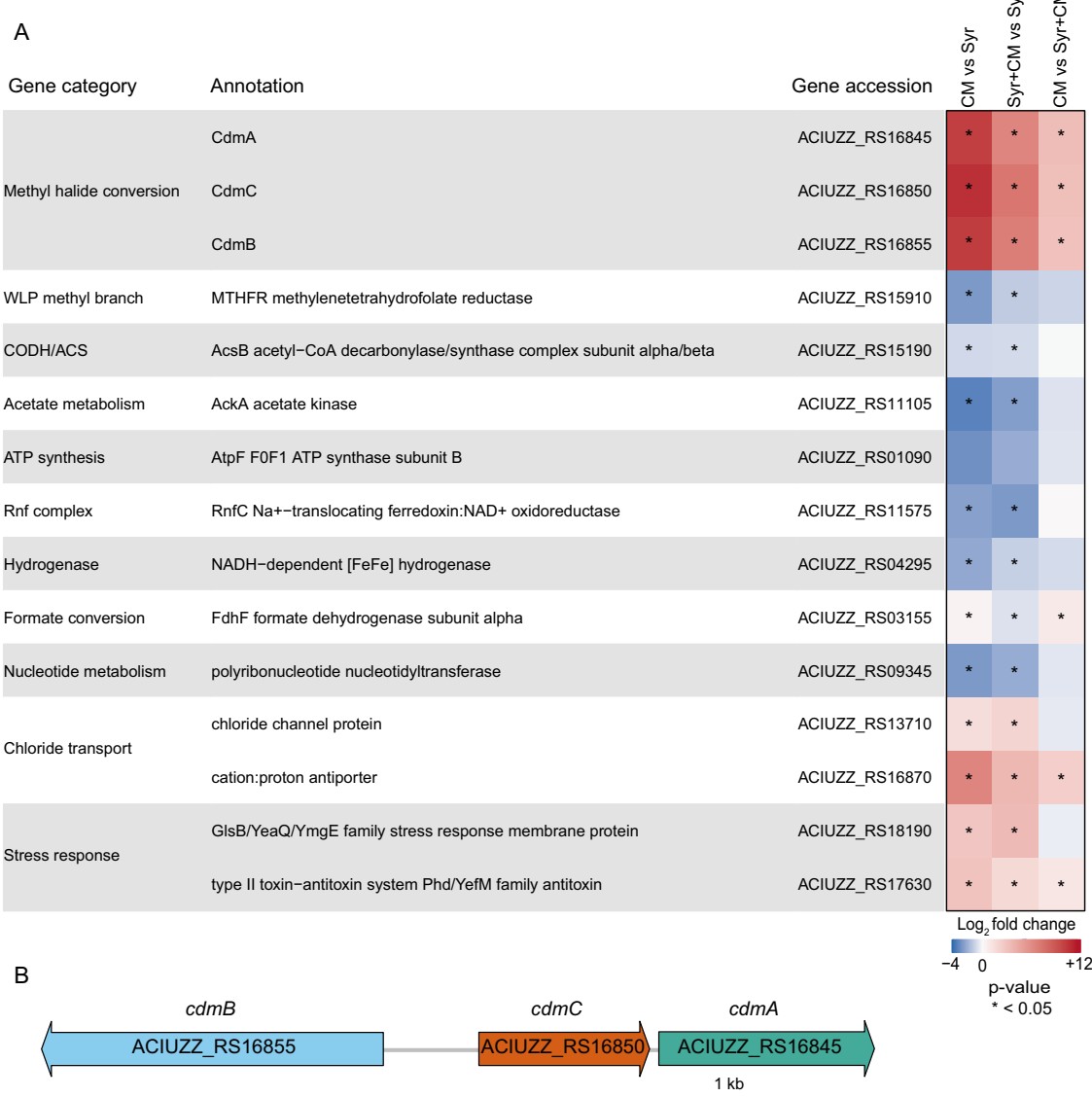

**Fig. 1 | Identification of the methyltransferase system involved in chloromethane metabolism in _A. dehalogenans_ by comparative transcriptomics.** Comparison of gene expression during growth of _A. dehalogenans_ on CM, syringate (Syr) or both substrates (**A**) and _cdmBCA_ gene cluster (**B**). **A** Heatmap illustrating the gene expression profile of _A. dehalogenans_ grown on CM, syringate (Syr), or a combination of both substrates (_n_ = 3 biological replicates). Log₂ fold change values in blue over white to red (−4 over 0 to +12) and _p_-values (<0.05, *) are shown for comparison between CM, syringate, and their combination (Supplementary

Fig. 1A, C, E). _P_-values were calculated using the two-sided Wald test (DESeq2) and adjusted for multiple testing with the Benjamini-Hochberg method. Complete RNA sequencing data are available in Supplementary Data 1. WLP Wood–Ljungdahl pathway, CODH/ACS: CO dehydrogenase/acetyl-CoA synthase, Rnf complex: Ferredoxin:NAD⁺ oxidoreductase. **B**, _cdmBCA_ gene cluster consisting of _cdmB_ (ACIUZZ_RS16855, encoding MTI), _cdmC_ (ACIUZZ_RS16850, encoding CP), and _cdmA_ (ACIUZZ_RS16845, encoding MTII).

reactive, followed by brominated and then chlorinated compounds. A specificity of the Cdm system for haloalkanes was clearly evident from the fact that MACs, methylated/methoxylated alcohols, dimethyl disulphide, and methylamines, were not demethylated by the Cdm system (Source Data: Table 1).

Taken together, our findings suggest that CdmB is required for Co(I)-CdmC reactivity with CM, and suggest that binding of CdmB to CdmC is necessary for CM conversion. However, the Cdm system also transforms DCM, iodomethane and longer-chain haloalkanes, including bromoethane, 1-bromopropane and 1-bromobutane, but not 1-chloropropane and 1-bromopentane (Table 1). This reaction could also be followed by formation of Co(III)-CdmC (Fig. 2B). The spectral characteristics of Co(III)-CdmC varied with haloalkane chain length, with a blue shift observed for longer-chain haloalkanes and a red shift for DCM (Supplementary Fig. 6). This indicates formation of

alternative reaction products to CH₃-Co(III) from CM, most probably CH₃-(CH₂)ₓ-Co(III) or Cl-CH₂-Co(III), respectively. Kinetic parameters for CdmB were determined for CM and DCM as substrates (Supplementary Fig. 7). CdmB exhibits a $K_m$ value of 2.9 mM for CM with a $V_{max}$ value of 4.8 μmol·min⁻¹·mg⁻¹ ($k_{cat}$ 3.8 s⁻¹). For DCM, a $K_m$ value of 68.7 mM and a $V_{max}$ value of 4.8 μmol·min⁻¹·mg⁻¹ ($k_{cat}$ 3.9 s⁻¹) were determined. The approximately 24-fold lower $K_m$ of CdmB for CM suggests that DCM may be recognised and bound less efficiently than CM, as expected since CM, unlike DCM, is a growth substrate of _A. dehalogenans_. Indeed, growth experiments involving the addition of DCM or long-chain haloalkanes to _A. dehalogenans_ cultures grown on either CM or syringate as growth substrate (Supplementary Fig. 6B) revealed that DCM actually inhibits growth on CM, suggesting specific inhibition of the Cdm system, possibly due to irreversible binding of DCM to the CdmC corrinoid. Longer-chain haloalkanes also appear to

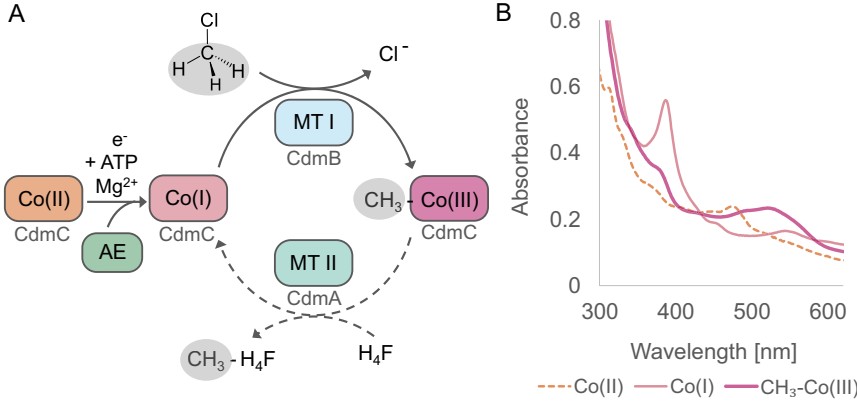

**Fig. 2 | CM dehalogenation and methyl transfer catalysed by Cdm proteins.**
Proposed reaction mechanism of CM dehalogenation by the CdmBC system from
*A. dehalogenans* (**A**) based on characteristic changes in cobalamin states of the CP
CdmC reflected in its UV-vis spectra over the reaction course (**B**). Formation of CH$_3$-
H$_4$F from CH$_3$-Co(III)-CdmC is then catalysed by CdmA (dashed arrow), as
suggested by the AlphaFold model of CdmA (Supplementary Fig. 5). UV-vis spectra
show the inactive Co(II) state (orange dashed line), the active Co(I) state (light pink
line) after addition of AE, ATP, Mg$^{2+}$ and electron donor to CdmC, and the CH$_3$-
Co(III) state (dark pink line) of CdmC after subsequent addition of CdmB and CM.
Source data are provided as a Source Data file.

## Table 1 | CdmB activity assay with halogenated alkanes

| Group | Substrate | CdmB-dependent Co(III)-CdmC formation | Specific activity of CdmB ($\mu$mol min$^{-1}$ mg$^{-1}$) |
|---|---|---|---|
| **X-C1** | CM | + | 3.9 ± 0.5 |
| | DCM | + | 3.2 ± 0.3 |
| | iodomethane | a | a |
| **X-C2** | iodoethane | + | b |
| | bromoethane | + | 0.6 ± 0.1 b |
| **X-C3** | 1-chloropropane | - | N.D. |
| | 1-bromopropane | + | 3.4 ± 0.2 |
| **X-C4** | 1-bromobutane | + | 1.7 ± 0.1 |
| **X-C5** | 1-bromopentane | - | N.D. |

Specific activities of CdmB were measured at 20 °C instead of the *A. dehalogenans* temperature
optimum of 28 °C to allow monitoring of the fast reaction. Values for CdmB-specific activity are
shown as mean (*n* = 3 technical replicates) ± standard deviation. Iodomethane, iodoethane and
bromoethane lead to Co(III)-CdmC formation independent of CdmB addition. Source data are
provided as a Source Data file.
X halogen, N.D. not detectable.
a It was not possible to determine the CdmB-dependent specific activity for iodomethane due to
the very rapid reaction rate.
b To account for the spontaneous reaction of iodoethane and bromoethane with Co(I)-CdmC in
the absence of CdmB, Co(III)-CdmC formation rates measured without CdmB were subtracted
from the activity measured with CdmB present in the reaction mixture. No significant CdmB-
specific activity could be detected for iodoethane.

exert a growth-inhibitory effect, likely due to their spontaneous reaction with CPs in general[38], potentially resulting in widespread inactivation of their methyltransferase systems. This further suggests that the reaction of DCM and longer-chain haloalkanes with the Cdm system is unlikely to be a physiologically relevant reaction. Further studies are needed to elucidate the reaction mechanism of the Cdm system and confirm its inhibition by CdmC inactivation.

### The methyltransferase CdmB features a substrate channel network that directs small haloalkanes to the cobalt ion within CdmC

To date, only a few microbial corrinoid-dependent MTIs have been structurally characterised[39–42], but none of these use haloalkanes as a substrate. To the best of our knowledge, we obtained the first crystal structure of a haloalkane-specific methyltransferase, a significant breakthrough in understanding the molecular basis of this important

reaction. The crystal structure of isolated CdmB was refined to a resolution of 1.57 Å. Results from the DALI server analysis indicate that its closest structural homologue is the human uroporphyrinogen III decarboxylase UroD (Supplementary Fig. 8) with a Z-score of 26.4 (root mean square deviation of 2.9 Å, with 308 residues aligned and 16% identity).

CdmB organises as a homodimer constituted of a TIM barrel harbouring several extensions between the first three β-sheets (Fig. 3A, B). The extensions mediate dimerisation and an internal hydrophobic core that exhibits a tunnelling system connecting the protein surface to the central TIM barrel cavity (Fig. 3C, D). The central cavity proposed to harbour the catalytic centre is formed mainly by aromatic residues with no noticeable metals. Despite being added to the protein before crystallisation, no zinc ion was observed in the structure.

To understand how halogenated substrates diffuse within the enzyme and locate the suggested active site, CdmB crystals were soaked with a high concentration of DCM or iodomethane. The use of iodomethane was particularly insightful as it closely mimics CM and its iodine atom could be precisely localised via X-ray fluorescence (XRF) (Supplementary Table 1). Both haloalkanes were detected in the hydrophobic channels with several differentially populated sites depending on the substrate (Fig. 3E, F, Supplementary Fig. 9). AlphaFold 3 modelling suggests an overall conservation of the hydrophobic tunnelling system among CdmB-like enzyme homologues (Supplementary Fig. 10), despite poor conservation of the residues composing the tunnel (Fig. 3G), implying the possibility of fine-tuning to enhance substrate selectivity.

One of the sites containing CH$_3$I is exposed to the solvent in the central cleft at a position that would ideally be in front of the Co(I)-B$_{12}$ bound to CdmC. An AlphaFold 3-predicted structure of the Cdm(B$_2$C$_2$) complex suggests that CdmC anchors to CdmB at two points (Supplementary Fig. 11), with the CdmC Rossmann fold domain containing the cobalamin protruding into the central hydrophobic cavity of CdmB (Fig. 4A). This model supports the hypothesis that the pre-catalytic CdmBC complex positions the Co(I) near the exit point of the hydrophobic channelling system. However, significant steric clashes occur when B$_{12}$ is superimposed, preventing a definitive conclusion on the corrinoid positioning in CdmB (Fig. 4B). This suggests that binding of active Co(I)-CdmC to CdmB involves structural rearrangements, allowing the corrinoid to engage deeper in the hydrophobic cleft to activate and shield the active Co(I) from the solvent, in a movement putatively analogous to what was described in the B$_{12}$-dependent methyl transfer to the CODH/ACS in *Clostridium autoethanogenum*[43]. Formation of such a conformational lock in the main hydrophobic

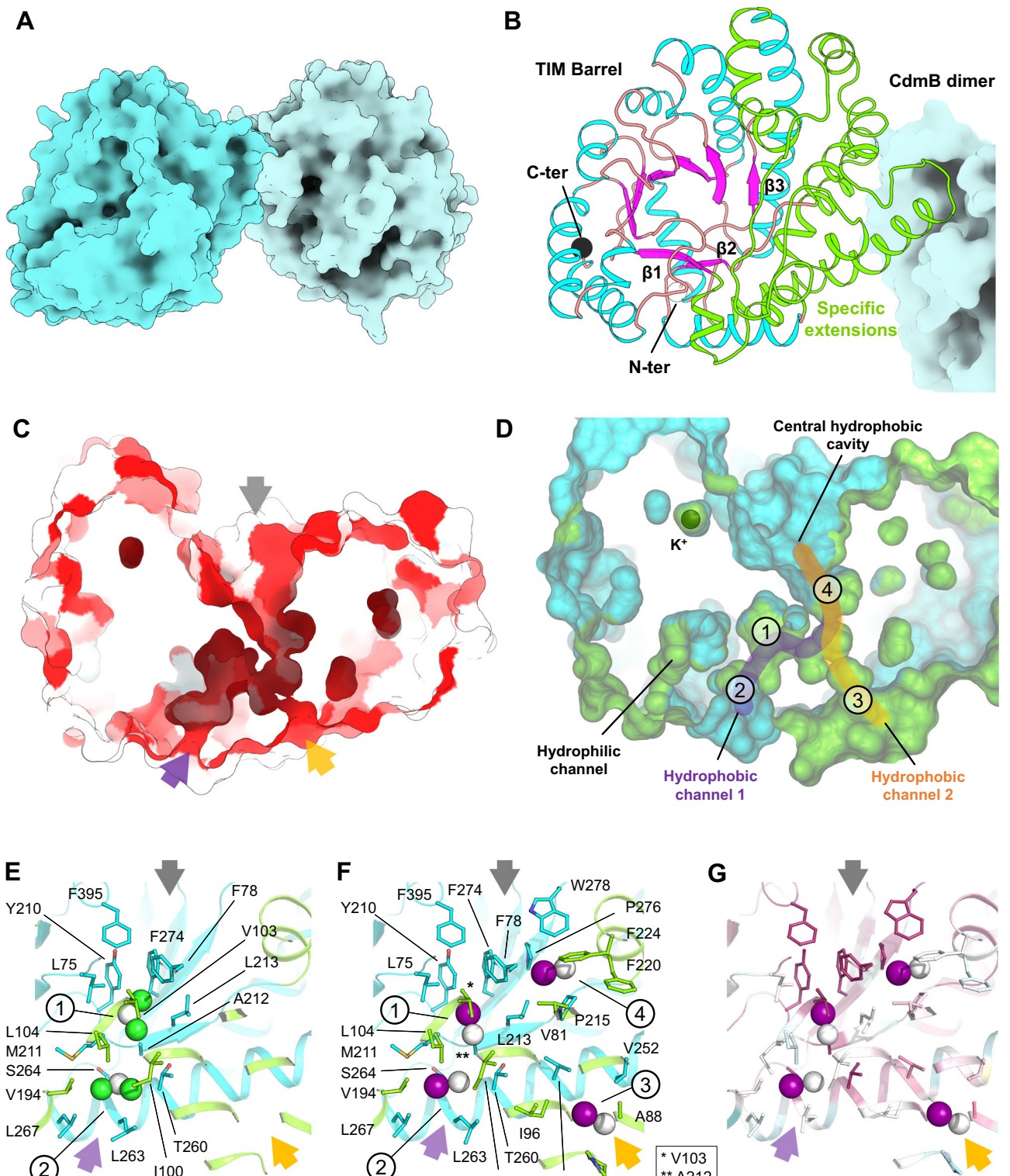

**Fig. 3 | CdmB binds haloalkanes through a dedicated hydrophobic network.**
**A** Overview of homodimeric CdmB apoprotein shown as a surface representation, with each monomer differently coloured. **B** Close-up view of one CdmB monomer displayed in cartoon style, and the second monomer displayed as a cyan surface. **C** CdmB apoprotein cut-through view displayed as a surface. The colour code corresponds to the hydrophobicity of the protein from hydrophilic (white) to hydrophobic (red), highlighting internal hydrophobic channelling systems. **D** CdmB apoprotein cut-through view displayed as a surface coloured in blue, with the specific extensions in green. Internal channels are shown, and halogenated substrate binding sites 1–4 are indicated. **E–G** Close-up views of the ligand binding

site in CdmB structures in complex with DCM (**E**) and CH₃I (**F**). CdmB is depicted in cartoon style as in (**B**), with ligands shown as spheres (chlorine and iodine coloured green and deep purple, respectively) and surrounding residues displayed as balls and sticks. Entry channels are indicated by arrows coloured purple, orange, and grey for hydrophobic channels 1, 2, and the central cavity, respectively. Based on occupancy (Supplementary Fig. 9), channel 1 is preferred. **G** CdmB in the same orientation as in (**E**) illustrates amino acid conservation, ranging from variable (dark cyan) to conserved (dark pink), as calculated from 16 sequences sharing at least 40% identity with CdmB (Supplementary Data 2).

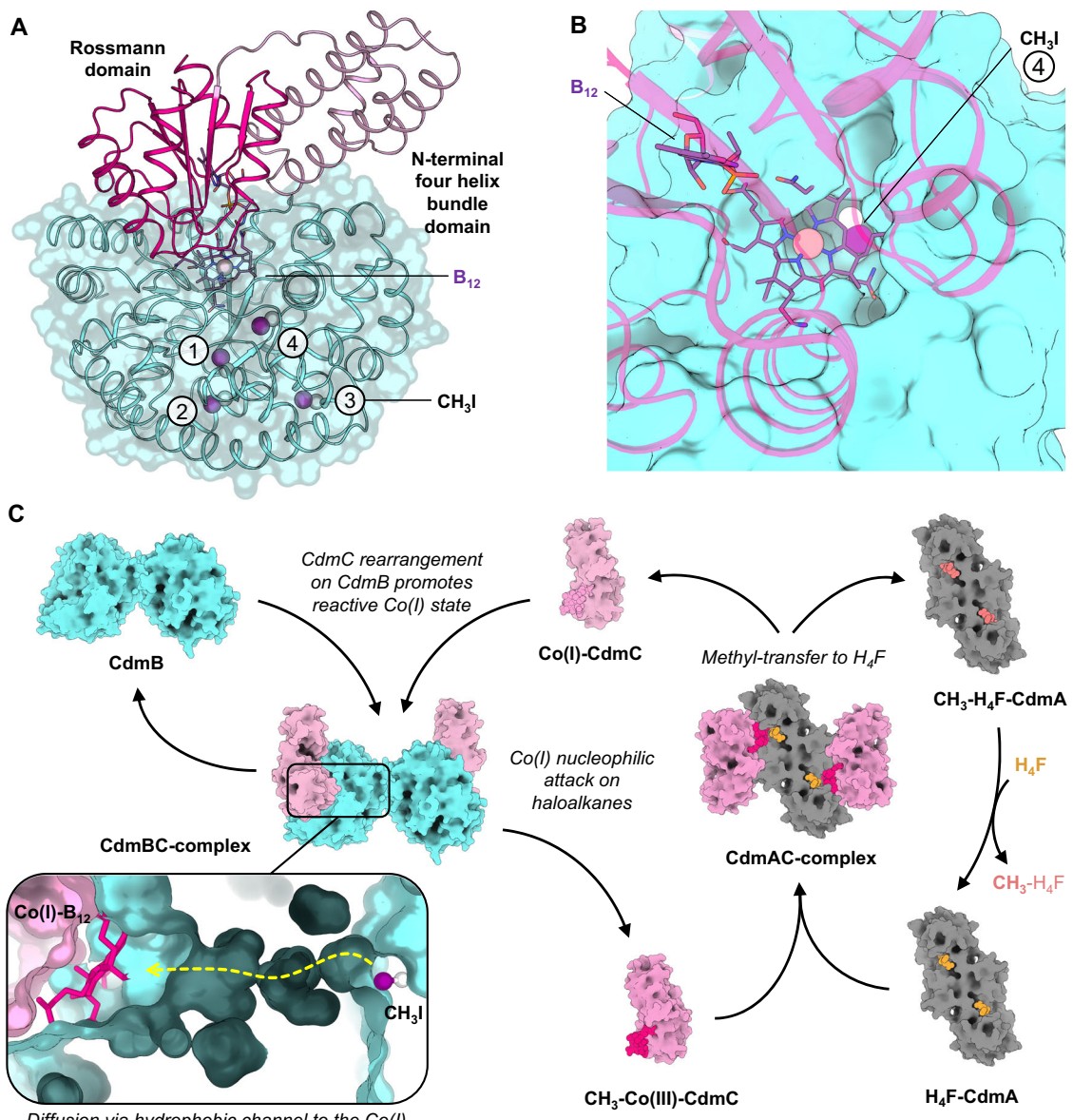

**Fig. 4 | Proposed reaction mechanism of the Cdm system. A** AlphaFold 3 model of the CdmBC complex, presenting both CdmC domains anchored to CdmB, visualised with a transparent surface. The $B_{12}$ position was modelled based on MetH (PDB 3BUL), and $CH_3I$ molecules were modelled based on the experimental structure. **B** Top view of (A) showing CdmB with a non-transparent surface, highlighting the steric clash between CdmB and the modelled $B_{12}$. $CH_3I$ from site 4 is visible at the bottom of the cavity. The cobalt-$CH_3$ distance of 10 Å in this model suggests a required conformational rearrangement of Co(I)-$B_{12}$-CdmC upon binding to CdmB to allow an adequate positioning of the reactive Co(I) in front of site 4 (Supplementary Fig. 11). **C** Proposed overall reaction mechanism of the Cdm system. During CdmBC assembly, the Rossmann domain of CdmC, which interacts with CdmB, undergoes a conformational change that enhances the reactivity of Co(I) and likely repels solvent from the hydrophobic cavity. CdmB, acting as a scaffold protein, selectively binds and directs the haloalkane substrate to the central cavity via a hydrophobic network in proximity to Co(I). Following the nucleophilic attack, $CH_3$-Co(III)-CdmC binds to CdmA loaded with $H_4F$ for the methyl transfer reaction. Finally, Co(I)-CdmC is regenerated, and CdmA releases $CH_3$-$H_4F$ to the central catabolic pathway. The structures of CdmC and CdmA were modelled using AlphaFold 3, with ligands superposed from PDB 4O1E ($CH_3$-$H_4F$ in CdmA), PDB 4O1F ($H_4F$ in CdmA) and PDB 3BUL ($B_{12}$ in CdmC).

cavity at the CdmBC interface would be ideal for nucleophilic attack of Co(I) on haloalkanes diffusing through the hydrophobic channel (Fig. 4C).

**The Cdm methyltransferase system exhibits an evolutionary history with homologues predominantly encoded by Bacillota and Asgardarchaeota**

Phylogenetic analysis of CdmB sheds light on the evolutionary history and distribution of the Cdm system (Fig. 5). CdmB belongs to the UroD superfamily, which includes *O*-demethylases as well as haloalkane dehalogenases. Proteins closely related to CdmB, such as MtvB[29],

OdmB[30] and VdmB[31] demethylate MACs (Fig. 5). CdmB, however, exhibits a clearly distinct specificity for haloalkanes, despite being most closely related to the archaeal *O*-demethylase MtoB[27,44] (31% sequence identity). In contrast, CmuA[45], MecE, and MecC[25,46], the only other known haloalkane-dehalogenating MTIs, appear to have diverged early within the UroD superfamily, further highlighting CdmB's evolutionary path. Cdm-like systems are present in Bacillota, with varying gene synteny in the *cdmBCA* gene cluster between the taxa (Supplementary Data 3). Screening genomes of haloalkane-degrading bacteria, such as *Formimonas warabiya*, *Desulfitobacterium chlororespirans*, and *Dehalobacter* spp. for MTIs with high

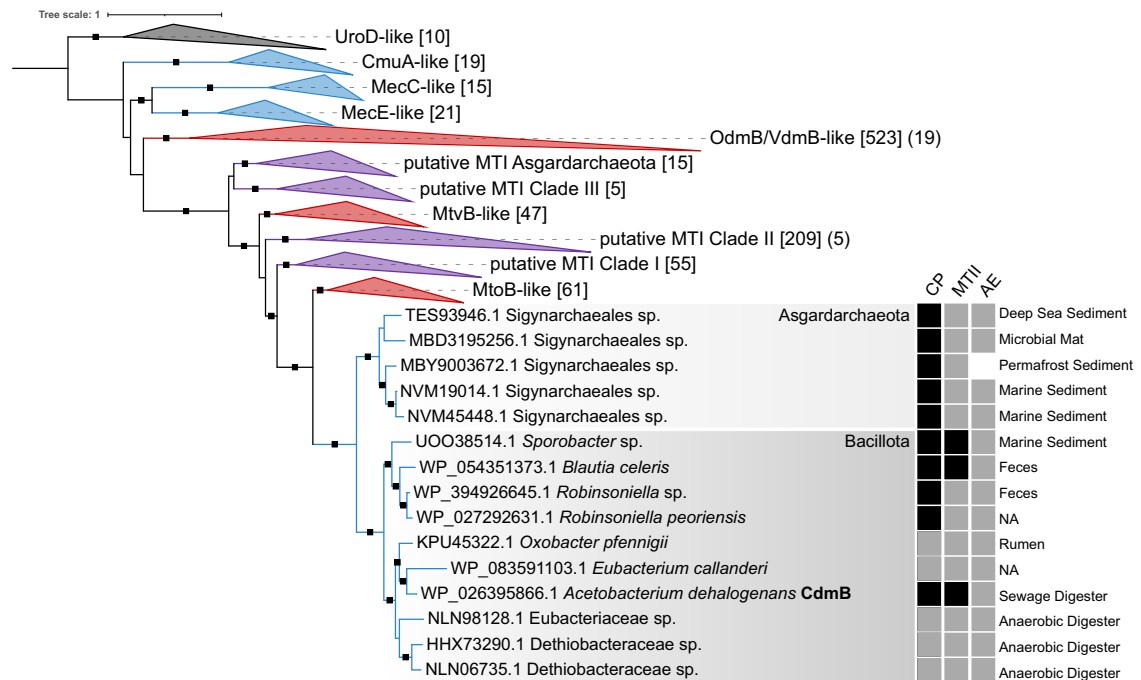

**Fig. 5 | Phylogenetic tree of CdmB, including Cdm system completeness and genomic isolation sources.** The maximum likelihood tree was rooted using the uroporphyrinogen decarboxylase UroD as an outgroup. Clades are colour-coded to represent MTIs that demethylate unknown substrates (purple), CM/DCM (blue), and MACs (red). Branch support is indicated by black squares, with ultrafast bootstrap and Shimodaira–Hasegawa test values above 95 and 80, respectively. Node labels show the total number of leaves and *A. dehalogenans* paralogs in square and round brackets, respectively. System completeness was assessed by checking the genomic presence of CP, MTIIs (CdmA or MtoA) and AE genes based on gene synteny up to five genes away from *cdmB* homologues (black squares) or presence in the genome using BLASTp (grey squares). MTIs of haloalkane-degrading bacteria with the highest sequence identity to CdmB were found to be scattered across uncharacterised clades within the UroD superfamily: Clade I: WP_148136635.1 from *Formimonas warabiya*, Clade II: WP_072772789.1 from *Desulfitobacterium chlororesipirans*, Clade III: WP_025205521.1, WP_015043224.1 from *Dehalobacter* spp. NA: not available. Source data are provided in Supplementary Data 3.

sequence identity to CdmB allowed identification of further proteins that might dehalogenate haloalkanes. These proteins are scattered across uncharacterised clades within the UroD superfamily. Closely related CdmB homologues are also found in Archaea, specifically in the Sigynarchaeales order within the Asgardarchaeota phylum, indicating that Archaea also possess the genetic potential for haloalkane conversion. The co-localisation of *mecE*-like genes with the *cdm*-genes in these microorganisms suggests that they may utilise DCM through a combination of methyltransferases. Positioning of the MTII CdmA (Supplementary Fig. 12) shows that CdmA is closely related to MtqA from *Eubacterium limosum* (59% sequence identity), the MTII involved in quaternary amine conversion[47–49]. Notably, several closely related CdmA paralogs are present in *Acetobacterium* species and appear to be associated with methyltransferase systems for different methyl compounds.

## Discussion

In this study, we identified and characterised a CM dehalogenation system (Cdm) responsible for CM conversion in *A. dehalogenans* and other anaerobic microorganisms. The *cdmBCA* genes are highly upregulated in response to CM (Fig. 1), but a regulatory system has not yet been identified. Overall, regulation of methylated compound conversion by the 26 different methyltransferase systems in *A. dehalogenans,* as well as regulation of bacterial haloalkane degradation pathways, remains largely unexplored[50]. In addition, Cdm does not convert MACs, despite its close phylogenetic relationship to *O*-demethylases. The Cdm system consists of an MTI, a CP, and an MTII, similar to *O*-demethylase systems[27,29–32] but distinct from the Cmu system, the only previously characterised CM dehalogenation system with CmuA acting as MTI and CP and CmuB as MTII[45,51]. Unlike some *O*-demethylases, such as veratrole and vanillate *O*-demethylases[28], which contain zinc as a cofactor, zinc is not present in Cdm, as also previously observed for the *O*-demethylase of *Moorella thermoacetica*[29].

The CdmB enzyme exhibits a high specific activity of $3.9 \pm 0.5$ μmol min⁻¹ mg⁻¹ for CM (Table 1). In comparison, CmuA of *M. chloromethanicum* had a specific activity for CM of $1.6$ μmol min⁻¹ mg⁻¹[51], while various *O*-demethylases had specific activities ranging from 0.2 to 0.9 μmol min⁻¹ mg⁻¹[27,28] for MACs. Notably, the Cdm system can transform both CM and DCM, as well as longer-chain haloalkanes like 1-bromobutane, contrary to Cmu[52] or the DCM-dehalogenating methyltransferase system, Mec of *Dehalobacterium formicoaceticum*[25]. However, CM is the only haloalkane shown to support the growth of *A. dehalogenans*[21] so far. In contrast to the Cmu and Mec systems, monomeric haloalkane dehalogenases from aerobic bacteria such as *Rhodococcus erythropolis*, *Mycobacterium avium*, *Sphingomonas paucimobilis*, and *Xanthobacter autotrophicus*[53] can dehalogenate various haloalkanes hydrolytically, including iodoethane, bromoethane, 1-bromopropane, and 1-bromobutane, with specific activities ranging from 1.7 to 6.7 μmol min⁻¹ mg⁻¹, which is comparable to CdmB.

The $K_m$ value of CdmB with CM as a substrate of approximately 2.9 mM is comparable to the $K_m$ values reported for quaternary amine MTIs (between 0.4 and 11 mM)[47–49,54]. However, it is higher than the $K_m$ values reported for the *O*-demethylases MtvB (0.09 mM with vanillate as substrate)[29], OdmB (about 0.1 mM with vanillate as substrate)[55] and the trimethylamine MTI MttB (0.05 mM for trimethylamine)[41]. For DCM, previously characterised dehalogenases exhibit $K_m$ values between 0.02 and 0.05 mM and $V_{max}$ values of up to 5.8 μmol · min⁻¹ · mg⁻¹, with $k_{cat}$ values ranging from 0.6 to 3.3 s⁻¹[56]. The $V_{max}$ of CdmB with DCM (4.8 μmol·min⁻¹·mg⁻¹) is in the same range as for CM.

In contrast, its $K_m$ value for DCM is much higher (68.7 mM). The high $K_m$ value observed for DCM is consistent with values reported for other methyltransferase reactions with non-physiological substrates, such as MtvB with dicamba, for which the $K_m$ is approximately two orders of magnitude higher than that of the physiological substrate vanillate[29]. We hypothesise that the difference in affinity for CM and DCM is due to the CdmB protein's selective channelling system. UV-vis spectroscopy clearly showed that CdmC forms a Co(III) complex not only with CM and DCM, but also with some longer-chain haloalkanes (Table 1). This is supported by the shifted 520 nm peak of Co(III)-CdmC observed with DCM as well as with longer-chain haloalkanes (Supplementary Fig. 6A), and the inhibition of *A. dehalogenans* grown on CM, but not on syringate, after DCM addition (Supplementary Fig. 6B). This suggests that longer-chain haloalkanes are also dehalogenated and the alkyl chain becomes bound to the cobalt ion of the corrinoid cofactor. To our knowledge, such a reaction has not been previously described for any methyltransferase system. We propose that longer-chain haloalkanes, as well as DCM, inhibit the Cdm system and growth of *A. dehalogenans* by generating alkyl derivatives that cannot be further processed in metabolism, suggesting that the reaction of Cdm with these substrates is not physiologically relevant for growth support. The selectivity of the CdmB channelling system could help prevent such toxic side reactions.

The structural description of CdmB with DCM and iodomethane provides a molecular view of how specific hydrophobic channels guide halogenated substrates to the TIM barrel central cavity (Fig. 3). Based on this model and considering previous findings regarding the reaction mechanism of corrinoid-dependent methyltransferases[40,57], we propose the following hypothetical multistep reaction mechanism for methyl halide dehalogenation (Fig. 4C) in which (i) Co(I)-CdmC binds to CdmB and its Rossmann fold domain rearranges to enhance the reactivity of Co(I) within the cobalamin of CdmC locked on the central cavity of the TIM barrel of CdmB, repelling out water molecules to protect the Co(I); (ii) methyl halide substrates diffuse through the hydrophobic channel of CdmB to reach a hydrophobic pocket in the vicinity of the Co(I); (iii) nucleophilic attack of the Co(I) onto the carbon of the substrates provokes the liberation of the halogen and the covalent binding of the methyl group; (iv) $CH_3$-Co(III) formation leads to the disengagement of CdmC from CdmB, allowing the release of the halogen and the next reaction step with CdmA. According to this hypothetical scenario, CdmB would not actively participate in the catalysis as shown for other MTI systems (such as MtaB[32,40] and MtmB[39,58] for the respective methanol and monomethylamine activation in methanogenic systems). The protein is rather a scaffold to selectively guide the appropriate substrate to the right location for the subsequent cobalamin chemistry. The methyl group transferred to $H_4F$ after CM dehalogenation by the Cdm system (Fig. 2) would subsequently be channelled into the Wood–Ljungdahl pathway (WLP) and utilised for $CO_2$ and acetate production (Supplementary Figs. 1 and 13). Regeneration of the reducing equivalents ferredoxin and NADH, necessary for the WLP, is likely mediated by a bifurcating hydrogenase (HydABC) and the Rnf complex, as reported for *Acetobacterium woodii*[59] (Supplementary Fig. 13). However, the absence of a second hydrogenase aside from HydABC in *A. dehalogenans* raises questions about a potential coupling of formate dehydrogenation to hydrogen production via a hydrogen-dependent carbon dioxide reductase (HDCR) as observed in *A. woodii*[60].

Based on phylogenetic analysis of CdmB, we hypothesise that haloalkane-converting methyltransferases may have evolved multiple times within the UroD superfamily and could be more widespread than previously thought. The *cdmB* gene is found in various microorganisms inhabiting gastrointestinal tracts such as *Blautia*, *Robinsoniella* and *Eubacterium* species, suggesting they can convert CM and other haloalkanes using Cdm. Previously, CM has been detected in human breath and milk[61,62] and was found to be emitted by cattle[63],

highlighting its significant but understudied role in gastrointestinal systems. Moreover, we identified Sigynarchaeales species encoding Cdm homologues, which may contribute to CM degradation in marine sediments. The phylogenetic position of archaeal CdmB homologues raises the possibility of horizontal gene transfer between the two domains. However, while corresponding Archaea are mainly present in (marine) sediments, *cdm*-containing bacteria are primarily found in gut-associated niches and digester systems (Fig. 5). Despite their complex evolutionary history, the distribution of these genes emphasises the potential importance of the Cdm system for anaerobic haloalkane conversion in these environments. Our analysis also revealed CdmB homologues in bacterial strains known for haloalkane degradation, but with unknown functionality. Overall, our findings underscore the need for further investigation into the distribution and functional diversity of haloalkane-converting enzyme systems

In summary, our study reveals a methyltransferase system, Cdm, which specifically dehalogenates haloalkanes such as CM and substantially differs from the characterised CM dehalogenase Cmu in composition, activity, and substrate spectrum. The CdmB structure provides structural insights into a methyltransferase participating in haloalkane dehalogenation. In contrast to other characterised methyltransferases, CdmB likely serves as a substrate-selective scaffold, channelling substrates to the cobalamin cofactor via a hydrophobic network, rather than actively participating in catalysis. Despite the close relationship of CdmB to *O*-demethylases, this methyltransferase does not convert MACs and likely followed a distinct evolutionary history. Our results suggest that the Cdm system is likely exclusive to anaerobic microorganisms, since *cdmB*-like genes are encoded by anaerobic members of the Bacillota phylum and of the Asgardarchaeota phylum considered to comprise mainly anaerobic Archaea. The presence of *cdmB* genes in Asgardarchaeota, suggests that Archaea may also possess the genetic potential to convert haloalkanes, thereby expanding our understanding of microbial haloalkane metabolism.

## Methods

### Cultivation of *Acetobacterium dehalogenans*

*A. dehalogenans* strain MC (DSM 11527), recently reclassified as *Acetobacterium malicum* ssp. *dehalogenans*[64], was obtained from Dr Sandra Studenik (Friedrich Schiller University Jena), a former co-worker of Prof Gabriele Diekert, in whose group the strain was isolated and investigated[21,22,28,31]. The organism was cultivated in modified basal medium[21]. One litre of basal medium contained 0.1% (w/v) $NH_4Cl$, 3 ml 1 M potassium phosphate buffer (pH 7.5), 40 ml 0.5 M sodium phosphate buffer (pH 7.25), 0.01% (w/v) $MgSO_4 \times 7\,H_2O$ and 0.0001% (w/v) resazurin, 1 ml vitamin solution (Wolin 1000×)[65], 2 ml trace element solution (500× solution: $5\,g\,l^{-1}$ nitrilotriacetic acid, $100\,ml\,l^{-1}$ 0.5 M NaOH, $6.2\,g\,l^{-1}$ $MgSO_4 \cdot 7\,H_2O$, $5.4\,g\,l^{-1}$ $MnCl_2 \cdot 4\,H_2O$, $10\,g\,l^{-1}$ NaCl, $1\,g\,l^{-1}$ $FeSO_4 \cdot 7\,H_2O$, $1.7\,g\,l^{-1}$ $CoCl_2 \cdot 6\,H_2O$, $1.3\,g\,l^{-1}$ $CaCl_2 \cdot 2\,H_2O$, $0.5\,g\,l^{-1}$ $CuCl_2 \cdot 2\,H_2O$, $1.5\,g\,l^{-1}$ $ZnCl_2$, $0.1\,g\,l^{-1}$ $KAl(SO_4)_2$, $0.11\,g\,l^{-1}$ $Na_2MoO_4 \cdot 2\,H_2O$, $0.2\,g\,l^{-1}$ $NiCl_2$, $0.021\,g\,l^{-1}$ $Na_2SeO_3 \cdot 5\,H_2O$), and 0.2% (w/v) yeast extract. L-cysteine ($0.5\,g\,l^{-1}$) replaced $Na_2S$ at a 28-fold reduced concentration, and a 4-fold reduced concentration of $K_2CO_3$ ($2.5\,g\,l^{-1}$) was used instead of $NaHCO_3$. A volume of 50 ml medium was added to 120 ml serum bottles. The medium was sparged with $N_2$:$CO_2$ in an 80:20 ratio before autoclaving. Syringate (0.5 mmol, final concentration 10 mM), CM (1.76 mmol or 0.65 mmol), or a combination of both (0.25 mmol, final concentration 5 mM, syringate plus either 1.76 mmol or 0.65 mmol CM) were used as carbon sources. For an addition of 0.65 mmol CM to the culture, CM was added every 24 h over a five-day period, with 4 ml (0.16 mmol) added on the first day and 3 ml (0.12 mmol) on each subsequent day. For an addition of 1.76 mmol CM to the culture, CM was added three times a day over a five-day period, starting with 4 ml (0.16 mmol), followed by 3 ml (0.12 mmol) at each subsequent addition. The volume of CM added was converted to mmol

**Table 2 | HPLC-MS parameters**

| Compound | Precursor ion | Product ion | Collision energy [V] | Fragmentor voltage [V] | Cell accelerator voltage [V] | Dwell time [ms] | Polarity |
|---|---|---|---|---|---|---|---|
| Syringate | 197 | 182 | 11 | 380 | 5 | 150 | Negative |
|  |  | 123 | 24 | 380 | 5 | 150 |  |
| DHMB | 183 | 167.8 | 12 | 380 | 5 | 150 | Negative |
|  |  | 124 | 19 | 380 | 5 | 150 |  |
| Gallic acid | 169 | 129 | 13 | 380 | 5 | 150 | Negative |
|  |  | 79.1 | 25 | 380 | 5 | 150 |  |
| Acetate-3NPH | 194 | 152 | 10 | 100 | 5 | 200 | Negative |
|  |  | 137 | 15 | 100 | 5 | 200 |  |

*DHMB* 3,4-dihydroxy-5-methoxybenzoic acid, *3NPH* 3-Nitrophenyl-hydrazine.

using the ideal gas equation and the molar volume of gas at 25 °C (24.46 l mol$^{-1}$). Cultures were incubated at 28 °C. Growth was followed by measuring the OD at 578 nm, and aliquots for HPLC-MS analysis and chloride assay were sampled.

Growth experiments with DCM and the longer-chain haloalkanes bromoethane, 1-bromopropane and 1-bromobutane were performed in triplicate to investigate inhibitory effects of DCM and longer-chain haloalkanes on Cdm and on the syringate methyltransferase system. DCM and longer-chain haloalkanes were added at a final amount (concentration) of 0.25 mmol (5 mM) to *A. dehalogenans* grown with either 0.25 mmol (6 ml) CM, 0.5 mmol (10 mM) syringate, or no supplement, in 120 ml bottles containing 50 ml of medium. The cultures were incubated at 28 °C, and the final OD at 578 nm was measured (Ultrospec 2100 pro UV/Vis Spectrophotometer, Biochrom).

### Analytical chemistry

The chloride assay was performed as described previously[66], with a few modifications. The final concentrations of HClO$_4$ and Fe(NO$_3$)$_3$ · 9 H$_2$O solutions used in the assay were 5.53 M and 20 mM, respectively, and these were prepared in ultrapure water prior to use as a reagent mix. Liquid culture samples were centrifuged for 15 min at 20,000 × g, after which 10 µl of the sample was mixed with 80 µl of the reagent mix. NaCl with final concentrations ranging from 0 to 10 mM was used for calibration. Measurements were performed in technical duplicates at 340 nm using a microplate reader (Synergy H1, BioTek) and Gen5 software (version 3.12).

Gas chromatography-flame ionisation detector (GC-FID) measurements were performed using an SRI 8610C instrument (SRI Instruments Europe GmbH) and PeakSimple 4.54. CO$_2$ and hydrogen (H$_2$) levels were monitored for *A. dehalogenans* grown on CM and syringate plus CM. During growth experiments, CO$_2$ levels increased over the incubation period: 0.44 mmol (4.9 mM) with 1.76 mmol CM, 0.34 mmol (3.7 mM) with 0.25 mmol (5 mM) syringate plus 0.65 mmol CM, 0.26 mmol (2.9 mM) with 0.65 mmol CM and 0.26 mmol (2.9 mM) with 0.25 mM (5 mM) syringate plus 1.76 mmol CM. After 56 h of incubation, H$_2$ levels between 3.25 µmol (0.037 mM) and 5.41 µmol (0.061 mM) were detected for all incubations.

Quantitative determination of syringate, 3,4-dihydroxy-5-methoxybenzoic acid, and gallic acid was performed by LC-MS/MS using filtered liquid samples from *A. dehalogenans* cultures (4 mm syringe filters, 0.2 µm particle size, Phenomenex). Chromatographic separation was performed on an Agilent Infinity II 1290 HPLC system using a Kinetex EVO C18 column (50 × 2.1 mm, 3 µm particle size, 100 Å pore size, Phenomenex) connected to a guard column of similar specificity (20 × 2.1 mm, 3 µm particle size, Phenomoenex), at a constant flow rate of 0.2 ml min$^{-1}$ with mobile phase A (0.1% formic acid in water) and phase B (0.2% formic acid in methanol) (Honeywell), at 25 °C. The injection volume was 1 µl. The profile of the mobile phase consisted of the following steps and linear gradients: 0–1.5 min constant at 5% B; 1.5–3.5 min from 5% to 100% B; 3.5–4.5 min constant at 100% B;

4.5–4.6 min from 100% to 5% B; 4.6–8 min constant at 0% B. An Agilent 6495 mass spectrometer was used in negative mode with an electrospray ionisation source and the following conditions: ESI spray voltage 2000 V, nozzle voltage 500 V, sheath gas 260 °C at 10 l min$^{-1}$, nebuliser pressure 35 PSIG and drying gas 100 °C at 13 l min$^{-1}$. For determination of acetate, 10 µl aliquots of sample and calibrants were sequentially mixed with 10 µl of both 250 mM 3-Nitrophenyl-hydrazine in 50% Methanol (LC-MS grade), and 7.5% pyridine in 75% Methanol (LC-MS grade) in a 96-well plate. After sealing the plate (HeatSealer, Eppendorf), the mix was incubated in a thermo block at 30 °C for 30 min, and centrifuged at room temperature (5 min, 8944 × g) before analysis. All solvents were freshly prepared on the day of the experiment. Quantitative determination of the derivatisation products was performed using LC-MS/MS. The chromatographic separation was performed on an Agilent Infinity II 1290 HPLC system using a Kinetex EVO C18 column (100 × 2.1 mm, 3 µm particle size, 100 Å pore size, Phenomenex) connected to a guard column of similar specificity (20 × 2.1 mm, 3 µm particle size, Phenomoenex) at a constant flow rate of 0.2 ml/min with mobile phase A being 0.1% formic acid in water and phase B being 0.1% formic acid in methanol (Honeywell, Morristown, New Jersey, USA) at 25 °C. The injection volume was 1 µl. The profile of the mobile phase consisted of the following steps and linear gradients: 0 – 3 min from 0% to 100% B; 3–4 min constant at 100% B; 4–4.1 min from 100% to 0% B; 4.1–8 min constant at 0% B. An Agilent 6470 mass spectrometer was used in negative mode with an electrospray ionisation source and the following conditions: ESI spray voltage 4500 V, nozzle voltage 500 V, sheath gas 300 °C at 11 l/min, nebuliser pressure 45 psig, and drying gas 170 °C at 5 l/min. Compounds were identified based on their mass transition and retention time compared to standards. Chromatograms were integrated using MassHunter software (version 10.0; Agilent, Santa Clara, CA, USA). Absolute concentrations were determined based on an external Standard curve. Mass transitions, collision energies, Cell accelerator voltages, and Dwell times have been optimised using chemically pure standards (Table 2).

### RNA isolation from *A. dehalogenans* and sequencing

For RNA extraction, *A. dehalogenans* was cultivated in 600 ml bottles containing 250 ml medium, with 2.04 mmol CM or 2.5 mmol (10 mM) syringate. For cultures with both syringate plus CM, amounts (concentrations) of 1.25 mmol (5 mM) syringate and 1.51 mmol CM were used. *A. dehalogenans* cells were harvested in the exponential phase after 48 h of incubation (cultures with 2.5 mmol (10 mM) syringate: OD$_{578}$ 0.384–0.452, cultures with 2.04 mmol CM: OD$_{578}$ 0.119–0.137 and cultures with 1.25 mmol (5 mM) syringate and 1.51 mmol CM: OD$_{578}$ 0.330–0.351) at 10,000 × g for 15 min at 4 °C. Cell pellets were frozen and stored at −80 °C until RNA isolation. RNA isolation was performed with the RNeasy® PowerSoil® Total RNA Kit (QIAGEN) according to the manufacturer's instructions. RNA samples (triplicates for each condition) were treated with DNA-free™ Kit DNase Treatment & Removal (Thermo Fisher Scientific) according to the manufacturer's

instructions. The quantity and quality of RNA were checked with a fragment analyser (Agilent). RNA quality numbers (RQN) were between 6.2 and 9.0. RNA sequencing library construction, rRNA depletion and RNA sequencing were performed by Genewiz (Leipzig, Germany). In short, RNA sequencing libraries were prepared using the NEBNext Ultra II RNA Library Prep Kit for Illumina following the manufacturer's instructions. Sequencing libraries were multiplexed and loaded on the flow cell of the Illumina NovaSeq 6000 instrument according to the manufacturer's instructions. Samples were sequenced using a 2 × 150 Pair-End configuration v1.5. Image analysis and base calling were conducted by the NovaSeq Control Software v1.7 on the NovaSeq instrument. The 2 × 150 bp paired-end reads were trimmed with trim_galore v0.6.7[67] using Cutadapt v4.1[68] and FastQC v0.11.9[69] in 2-colour and paired-end mode. Trimmed reads were aligned to the reference genome of *Acetobacterium malicum* subsp. *dehalogenans* DSM 11527 (CP174122.1, available from NCBI) using Bowtie2 v2.5.0[70] with the parameters '--very-sensitive' and '-k30'. Mapped reads were assigned to "gene" features using featureCounts (subread v2.0.3)[71] with the parameters '--countReadPairs -M --fraction'. Differential gene expression analysis was done with DESeq2 v1.38.0[72]. *P*-values were calculated using the two-sided Wald test (DESeq2) and adjusted for multiple testing with the Benjamini-Hochberg method.

## Heterologous protein production of CdmC, CdmB, and the activating enzyme

Genes encoding MTI CdmB (ACIUZZ_RS16855), corrinoid protein (CP) CdmC (ACIUZZ_RS16850), and CdmC-activating enzyme (AE) (ACIUZZ_RS15945) were amplified from total *A. dehalogenans* DNA with primers 6855fw/6855rev, 6850fw/6850rev, and 5945fw/5945rev (Table 3).

For cloning genes in the expression vector pET-30a (Novagen), an N-terminal StrepII tag was inserted via the reverse primer for ACIUZZ_RS16855 and a C-terminal StrepII tag via the forward primer for ACIUZZ_RS16850 and ACIUZZ_RS15945. Primers included an appended gene-flanking sequence region that was used for Gibson Assembly[73]. For vector linearization and amplification, primers pET30a_fw_JB and pET30a_rev_JB were used (Table 3). Primer design was performed manually in Benchling, and primers were ordered from Microsynth AG. PCRs were performed with Q5® High-Fidelity 2× Master Mix (NEB) according to the manufacturer's instructions. *E. coli* DH5α (NEB) was used for plasmid transformation. For production of CdmB (ACIUZZ_RS16855; WP_026395866), plasmid pET30a_ACIUZZ_RS16855 was used for transformation into *E. coli* BL21(DE3). For CdmC (ACIUZZ_RS16850; WP_026395867) and AE (ACIUZZ_RS15945; WP_026395886), plasmids pET30a_ACIUZZ_RS16850 and pET30a_A-CIUZZ_RS15945 were used for transformation into *E. coli* ArcticExpress (DE3) (Agilent). For production of CdmB, a preculture was prepared to inoculate LB medium containing 50 µg ml⁻¹ kanamycin and incubated at 37 °C for 16 h. For protein production of CdmC and AE, a preculture was prepared to inoculate ZYM-5052[74] medium containing 1× TE solution, supplemented with 50 µg ml⁻¹ kanamycin and 20 µg ml⁻¹

gentamycin, and incubated at 22 °C and 180 rpm for 24 h. Cells were harvested by centrifugation (15,000 × *g* for 10 min at 4 °C). For CdmC and AE, all further steps were performed anaerobically in an anaerobic chamber containing a gas atmosphere of N₂/H₂ at a ratio of 97:3% with anoxic buffers and solutions. Cell pellets were resuspended in 100 mM Tris-HCl buffer (pH 8) containing 150 mM NaCl and lysed by sonication (1 s pulse, 5 s pause, 40% amplitude; over 5 min). After removal of insoluble cell material by centrifugation (4690 × *g* for 20 min at 4 °C), proteins were purified by Strep-Tactin® Superflow® high-capacity affinity chromatography according to the manufacturer's instructions (IBA). For assessment of purity, sodium dodecyl sulphate-polyacrylamide gel electrophoresis (SDS-PAGE) was performed. Protein concentration was measured by the Bradford Protein Assay (Bio-Rad) according to the manufacturer's instructions and determined with a bovine serum albumin (BSA) protein standard. Reconstitution of CdmC with cobalamin was performed as described previously[27]. Refolding solution (1 ml per 0.5 mg protein) containing 50 mM Tris (pH 7.5), 3.5 M betaine HCl, 1 mM hydroxocobalamin HCl, and 10 mM DTT was prepared. The protein-containing solution was incubated under anaerobic conditions for 16 h at 4 °C in the dark with gentle stirring. For buffer exchange, 25 mM Tris-HCl (pH 7.5) containing 2 mM DTT and 10% (v/v) glycerol was used repeatedly with 10 kDa concentration units (Amicon Ultra-15 centrifugal filter units, Merck) until the cobalt-containing permeate appeared clear instead of red. Protein aliquots were stored anaerobically in 2 ml amber glass vials closed with airtight rubber stoppers.

## Enzyme activity assays

Enzyme activity assays were performed in anaerobic 400 µl quartz cuvettes as described previously[27]. All reactions were performed at 20 °C and at least in triplicate. To accommodate the high reactivity observed, the reaction temperature was reduced from the organism's optimal temperature of 28 °C to 20 °C. At 28 °C, decreasing the CdmB concentration resulted in its gradual deactivation, as evidenced by incomplete methylation of the Co(I) state to the Co(III) state, a limitation that could be alleviated by supplementing with additional CdmB. Gastight Hamilton syringes were used to add anoxic buffers and solutions. CdmB activity was determined in 35 mM Tris-HCl (pH 7.5) buffer with 70 mM KCl. The reconstituted Co(II)-CdmC at 0.6 mg ml⁻¹ final concentration (about 27.4 µM) was activated by adding 11 mM MgCl₂, 0.9 mM Ti(III)citrate, 2.2 mM ATP and 0.09 mg ml⁻¹ AE. Conversion to Co(I)-CdmC involving the appearance of a peak at 386 nm (gamma band) was followed using a Cary 60 UV–vis spectrophotometer (Agilent) and Cary WinUV 5.3 software. Conversion of Co(I)-CdmC to Co(III)-CdmC occurs due to coordination of a distinct group, such as a methyl group, to the cobalt ion and was either started by the addition of substrate or of CdmB at a final concentration of 0.01 mg ml⁻¹. When using CM, DCM, 1-bromopropane and 1-bromobutane as substrates, no conversion was observed without AE or CdmB. The following substrates were added: 150 µl CM (9.7 mM), 1 µl iodomethane (38.5 mM), 2 µl DCM (96.86 mM), 2 µl iodoethane

## Table 3 | Primer sequences

| Primer | Sequence (5' to 3' direction) |
|---|---|
| 6855fw | TAGTTATTGCTCAGCTTAGTACTTGCCATGTTTAAGGAT |
| 6855rev | GAGATATACATATGTGGAGTCATCCACAATTTGAGAAGTCAGCATCAAATCAAGTATTA |
| 6850fw | GTTATTGCTCAGCTTACTTCTCAAATTGTGGATGACTCCATGCTGAAACCAAATCTTGA |
| 6850rev | ACTTTAAGAAGGAGATATACATATGTTAGATTTAAATGTGTTAACACAAGCGTTAGGTG |
| 5945fw | GTTATTGCTCAGCTTACTTCTCAAATTGTGGATGACTCCATGCTGATTTCATTTCATTT |
| 5945rev | TAAGAAGGAGATATACATATGTCATCTTTGAATACTATTCGCGTTTTTTTCCCG |
| pET30a_fw_JB | CATATGTATATCTCCTTCTTAAAGTT |
| pET30a_rev_JB | GCTGAGCAATAACTAGCATA |

(63.6 mM), 5 μl bromoethane (157.2 mM), 2 μl 1-bromopropane (52.6 mM) and 2 μl 1-bromobutane (28.8 mM), 1-chloropropane (47.4 mM), 1-bromopentane (25.7 mM), 12 μl 3,4,5-trimethoxybenzoic acid (2.1 mM), 12 μl syringate (2.1 mM), 12 μl vanillate (2.1 mM), 12 μl 1-methoxy-2-propanol (2.1 mM) or 12 μl dimethyl disulphide (2.1 mM).

Formation of Co(III)-CdmC results in the disappearance of the peak at 386 nm and the appearance of a peak at about 520 nm[28]. To follow the entire dehalogenation/methyl transfer process, UV–vis spectra were recorded from 250–650 nm after sequential addition of CdmC, Ti(III)citrate, together with ATP, MgCl$_2$ and AE, substrate and CdmB. Methanol or mono- and trimethylamine at a concentration of 2.1 mM were used as negative controls. The decrease in absorption at 386 nm, caused by the formation of Co(III)-CdmC from Co(I)-CdmC, was followed with a Cary 60 UV–vis spectrophotometer (Agilent). A value of $\Delta\varepsilon_{386} = 21\,mM^{-1}\,cm^{-1}$[75] was used for calculations of enzyme activity. An increase in absorption at about 520 nm was visible in UV-vis spectra after the addition of CdmB and substrate, confirming Co(III)-CdmC formation. Assays were performed with and without the addition of CdmB. To test whether CM can react with free hydroxocobalamin, enzyme activity assays were performed in which 29 μM hydroxocobalamin was added instead of the corrinoid protein CdmC. To test if zinc is required for CdmB catalysis, 1 mM EDTA was added before CdmB addition, and the activity was compared to EDTA-free samples. Moreover, reactions with the addition of 1 mM EDTA and 1 mM ZnCl$_2$, as well as reactions with 1 mM EDTA and 2 mM ZnCl$_2$, were performed as described previously[76]. All enzyme activity assays were performed in triplicate except for the hydroxocobalamin assays, which were performed in duplicate. To generate Michaelis–Menten curves and calculate the kinetic parameters of CdmB, up to 10.2 mM CM and up to 474 mM DCM were used. Specific activity values and their standard deviations were calculated, and Michaelis–Menten curves were plotted using GraphPad Prism3. Nonlinear regression analysis was used for curve fitting and to obtain $V_{max}$ and $K_m$ values. The CM fraction in the gas phase of each cuvette was examined by gas chromatography using a Shimadzu GC-2030 equipped with a Barrier Ionisation Discharge Detector (BID) and a Carboxen®-1010 PLOT fused silica capillary column (30 m × 0.32 mm, Sigma-Aldrich) with Helium as a carrier gas (3.25 ml min$^{-1}$). The LabSolution software (version 5.124 SP1) was used for data acquisition. The oven temperature was set to 230 °C. 300 μl gas was taken from each cuvette and transferred into an 8 ml closed serum bottle filled with N$_2$ gas. Of this, 50 μl was injected into the Shimadzu GC-2030. Due to the GC detection limit, a 50 μl gas sample from cuvettes containing 5 and 10 μl of CM was injected directly into the GC without dilution. CM and DCM concentrations in the liquid phase were calculated using Henry´s law in combination with the ideal gas equation, using Henry´s constants of 0.0012[77] and 0.0044[78] mol m$^{-3}$ Pa$^{-1}$ for CM and DCM, respectively.

### Phylogenetic tree construction and homology analysis

To infer the phylogeny of CdmB and CdmA, homologous proteins of known functionality (including MtoB, MtvB, OdmB, VdmB, CmuA, MecC, MecE, UroD and 20 paralogs for CdmB and MtvA, MtqA, OdmD, VdmD, four bacterial MtaA, AcsE, MetH and 13 paralogs for CdmA) were used as queries for BLASTp on NCBI[79] against the non-redundant database to obtain homologues (Supplementary Data 3). Datasets were filtered to only include microbial proteins encoded by genomes present in the GTDB database v220[80] for taxonomic assignment. CD-Hit v4.8.1[81] was used to remove redundant sequences with a threshold of 98% and 97% sequence identity for CdmB and CdmA, respectively. Sequences were aligned under default parameters using Clustal Omega v1.2.4[82], and the alignments were trimmed manually using AliView v2021[83]. Phylogenies were inferred with IQ-TREE 2 v2.4.0[84] using ModelFinder[85] to determine LG+F+R9 and LG+I+R9 as the best-fitting model according to the Bayesian information criterion for CdmB and CdmA, respectively. Branch support was assessed using

1000 replicates of the Shimodaira–Hasegawa approximate likelihood-ratio test and ultrafast bootstraps. Both trees were visualised in iTOL v7[86]. System completeness was assessed manually by checking for the presence of CP, AE and MTII encoding genes up to five genes up or downstream of each *cdmB*-like gene. Additionally, BLASTp was used to screen for CdmC, CdmA, AE and MtoA (archaeal equivalent of CdmA) encoded in the respective genomes. Only hits with a similar sequence length as the query, e-values < 0.001 and bit scores >50 were considered. All BLASTp queries and system-completeness analyses are available in Supplementary Data 3. Phylogenetic trees and trimmed alignments are available on figshare (https://doi.org/10.6084/m9.figshare.29852144).

### Size-exclusion chromatography, crystallisation and soaks

Size-exclusion chromatography was performed immediately before crystallisation to remove potential aggregates. Purified CdmB was diluted with one volume of 25 mM Tris-HCl, pH 7.6, 10% (v/v) glycerol, 2 mM dithiothreitol, filtered, and injected on a HiLoad® 16/600 Superdex® 200 size-exclusion chromatography column (GE Healthcare), equilibrated with the same buffer. Chromatography was performed at a 1 ml min$^{-1}$ flow rate at room temperature. The protein eluted as a Gaussian peak and was concentrated using a 10-kDa cut-off centrifugal concentrator (nitrocellulose, Vivaspin from Sartorius). In case it was required for activity, ZnCl$_2$ was added at a final concentration of 0.1 mM to the protein before crystallisation. The protein was crystallised at 17.6 mg ml$^{-1}$. Protein concentration was estimated by the Bradford method using a BSA protein standard.

Initial screening was performed on a 96-well MRC 2-Drop polystyrene crystallisation plate (SWISSCI) at 20 °C. From the initial hit, crystallisation was further refined and performed aerobically at 20 °C using the sitting drop method on CombiClover® Jr crystallisation plates (Molecular Dimensions). The reservoir chamber was filled with 100 μl of the crystallisation condition: 50% (v/v) polyethylene glycol 200, 100 mM sodium potassium phosphate (pH 6.2) and 200 mM sodium chloride. The crystallisation drop was formed by spotting 0.9 μl of purified protein at 17.6 mg ml$^{-1}$ with 0.9 μl of precipitant. The crystals were either directly harvested or soaked in a crystallisation solution supplemented with 16% (v/v) DCM (CH$_2$Cl$_2$) for 6.2 min or 16% (v/v) iodomethane (CH$_3$I) for 4.8 min, before being frozen in liquid nitrogen. Anomalous data were collected at 7.100 keV from crystals soaked for 6.5 min in 16% (v/v) CH$_3$I.

### Structure determination, refinement, and model validation

Diffraction experiments used for the deposited models were performed at 100 K on the beamline FIP2-BM07 from the European Synchrotron Radiation Facility (ESRF), at a photon energy of 7.100 or 12.657 keV. Datasets were processed and scaled with *autoPROC*[87] (Version 1.0.5, Global Phasing Limited, Cambridge, UK), and all data except for the anomalous dataset collected at 7.100 keV were treated as anisotropic. The apoprotein structure was solved by molecular replacement with an AlphaFold 3[88] model by using PHASER[89] from the PHENIX package. The structures coming from the soaked crystals were solved via molecular replacement by using the apoprotein structure as a template. All models were then refined with COOT[90] and phenix.refine. Models were refined by applying translational-libration-screw (TLS) and by adding riding hydrogens. All models have been deposited without hydrogen atoms. The models were validated by the Mol-Probity server (http://molprobity.biochem.duke.edu). Figures were generated with PyMOL (V. 2.2.0, Schrödinger, LLC). Electron density maps of CdmB and omit maps of the ligands are shown in Supplementary Fig. 14.

### Reporting summary

Further information on research design is available in the Nature Portfolio Reporting Summary linked to this article.

## Data availability

Transcriptomics data were deposited under GenBank SRR32341242-SRR32341250. The raw sequencing files from this study are available at the NCBI Sequence Read Archive (SRA) under BioProject ID PRJNA1223667, including BioSample accessions SRR32341242-SRR32341250 [https://www.ncbi.nlm.nih.gov/bioproject/PRJNA1223667]. Plots and normalised count tables generated from this analysis are available on Zenodo [www.zenodo.org/records/19634717]. Phylogenetic trees and trimmed alignments are available on figshare [https://doi.org/10.6084/m9.figshare.29852144]. The CdmB models and their associated structure factors were deposited in the Protein Data Bank under the following accession codes: 9RUI (CdmB apo); 9RUL (CdmB soaked with DCM); and 9RUO (CdmB soaked with iodomethane). In the manuscript, we also refer to PDB codes 3BUL, 4O1E, and 4O1F. The HPLC-MS/MS raw data have been deposited on Figshare [https://doi.org/10.6084/m9.figshare.32089645]. The raw data from UV/Vis spectroscopy, $OD_{578}$ measurements, chloride assay, GC, and activity assay measurements, as well as the processed HPLC-MS data generated in this study, are provided in the Source Data file. Source data are provided with this paper.

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

## Acknowledgements

We thank Dr Sandra Studenik for providing us with *Acetobacterium dehalogenans* and the medium composition. We also thank Dr Doreen Meier (AG Anke Becker, Synmikro) and the Synmikro SAT (Screening and Automation) core facility for the use of their fragment analyser for QC of RNA samples. Furthermore, we thank the team of beamline BM07-FIP2 for their assistance during data collection, particularly Dr Sylvain Engilberge. We would like to thank Dr Miriam Kronen (AG Berg, University of Münster) for her assistance in developing a gas chromatography method for determining CM. O.N.L. and T.W. thank the Max Planck Institute for Marine Microbiology for continuous support. O.N.L. and T.W. acknowledge the French Biology/Health Panel Review Committee for the provision of synchrotron radiation beamtime at the ESRF (Grenoble, France) on beamline BM07-FIP2, supported by the French ANR PIA3 (France 2030) EquipEx+ project MAGNIFIX under grant agreement ANR-21-ESRE-0011.

## Author contributions

J.M.K., S.V. and T.W. conceived and designed the study. J.B. carried out most experiments except those of structural biology. Gel filtration, crystallisation, and crystal soaks were performed by O.N.L. O.N.L. and T.W. collected X-ray synchrotron data and solved the initial CdmB structure. T.W. refined and validated all structural models. T.W. and O.N.L. analysed the structural models. HPLC measurements were conducted by N.P. P.K. performed the transcriptomics analysis. J.M.K. and J.B. interpreted the omics data. L.K.R.H. conducted the phylogenetic analysis. J.M.K. and J.B. took the lead in writing the manuscript. Feedback from all authors was integrated into the manuscript. All authors discussed the results and validated the submitted version of the manuscript.

## Funding

J.M.K. was supported by the Deutsche Forschungsgemeinschaft (DFG, German Research Foundation) - Project number 532554430. L.K.R.H. acknowledges funding by the International Max Planck Research School Principles of Microbial Life. O.N.L. and T.W. were supported by the Max Planck Society. Open Access funding enabled and organized by Projekt DEAL.

## Competing interests

The authors declare no competing interests.
