## [Transparent Peer Review file · Nature Communications]

Identification and characterisation of an elusive bacterial enzyme system for chloromethane dehalogenation

Corresponding Author: Professor Julia Kurth

Version 0:

Reviewer comments:

Reviewer #2

(Remarks to the Author)

The manuscript entitled "Identification and characterisation of a long-elusive bacterial enzyme system for chloromethane dehalogenation" by Berhardt et al identified a gene cluster, *cdmBCA*, within *Acetobacterium dehalogenans* that encodes a novel corrinoid-dependent methyltransferase system, distinct from the characterized *Cmu* system used for chloromethane degradation in aerobic methylotrophs. The authors major findings are the identification of the *cdmBCA* gene cluster, the expression and purification of *CdmB* and *CdmC* and showed they could dehalogenate CM and DCM. They also solved the X-ray structure of *BdmB*. The work in this paper is thorough and thoughtfully presented; however, there are several shortcomings that likely prevent its publication in Nature Communications. I would suggest a more focused journal. My specific comments are below.

Comment 1. The authors report specific activities for *CdmB* towards CM and DCM as well as iodo, chloro, and bromo derivatives but fail to report *k_{cat}* or *K_m* values. These kinetic constants are required to understand substrate specificity and to gain insight into the mechanism of action. As it stands, Table 3 is mostly empty space with only a few specific activity values, providing little insight into the reaction in question.

Comment 2. While interesting, the *CdmB* structure is poorly presented in Figure 3 making it difficult for the reading to glean any relevant information. While examining the iodo derivatives binding could be interesting, what the authors observe are multiple binding sites which provide little insight into how substrate binds for catalysis.

Comment 3. On P. 22 the authors describe a catalytic mechanism, do not refer back to Figure 2, which makes it difficult to follow. What they describe does not appear to be supported by the data presented and is highly speculative. Additional data and a figure is needed to propose a reasonable mechanism.

Comment 4. *CdmA* was also upregulated in response to CM (according to Figure 1). Is there a reason it was not also heterologously expressed?

Minor comments

Within the introduction the authors fail to mention of the toxicity of CM and DCM to humans and other animals. While their focus is on ozone depletion and climate change (very important issues), acute toxicities of these compounds would further motivate and empathize their research. It may also be valuable to mention current methods for removing CM and DCM from the environment.

Line 191: Unnecessary close parenthesis present

Line 201: Is there a reason different concentrations for each potential substrate were used? How was each concentration decided on as the *K_m* values are unknown?

Line 242: "Diluted with one volume." What does "one volume" mean in this context?

Line 369: Is there a proposed reason large compounds are not substrates (size of active site?)?

Line 380-1: This hypothesis potentially clashes with the data presented in Table 3. If DCM is inhibiting the *Cdm* system, shouldn't specific activity be significantly lower for DCM than CM? If DCM is irreversibly binding *CdmC*, that would effectively decrease the concentration of *CdmC* in the reaction.

Line 382: Were both those that are substrates (bromopropane/bromobutane) and those which are not substrates

(bromopentane) inhibitory or just one of these groups?

Reviewer #3

(Remarks to the Author)

This paper by Berhardt et al. describes the identification and heterologous production of the proteins/enzymes involved in anaerobic dehalogenation/demethylation of chloromethane by *Acetobacterium dehalogenens* along with demonstration of the activity, substrate specificity, crystal structure, and substrate binding of the enzyme complex. I find this study to be a highly noteworthy contribution to the fields of dehalogenation research, anaerobic enzymology, and structural biology. The work is well-planned and executed and the work appears to be well-described for reproducibility purposes. It is a much needed and anticipated discovery that will prove to be a significant contribution to science.

I had only minor concerns/questions about the study:

1. The hypothesis or model that CdmB acts merely to channel and position CM for nucleophilic attack by Co(I)-CdmC is fascinating. This begs the question of whether CdmB and CdmC can function independently. The authors indicate in the results "Notably, iodomethane, iodoethane, and bromoethane reacted with CdmC even in the absence of CdmB, likely due to their high reactivity with cobalt." It was not clear to me whether Co(I)-CdmC could spontaneously react with CM in the absence of CdmB (I apologize if I missed that in the text). If CdmB merely guides and positions the CM for nucleophilic Co(I) attack, one would assume that there would be some low level of activity of CdmC methylation in the absence of CdmB. The spectra they showed seem to indicate that the Co(I) state of CdmC is relatively stable so perhaps the hydrophobic pocket created by CdmB to protect the Co(I) is not entirely needed.
2. In addition, I don't believe the authors tested the ability of CdmB to use CM to methylate Co(I)-free B12, which some MT1 enzymes can do with their specific substrates. Perhaps the authors have tried these experiments already and did not report them. If not, they seem to be quick and easy experiments to try that would help further elucidate the mechanism of this enzyme system and potentially support the hypothesis they proposed.

Version 1:

Reviewer comments:

Reviewer #2

(Remarks to the Author)

The authors address most of my comments so I suggest it be published in Nature Communications with some minor revisions.

Comment 1. As it stands, Table 3 is still mostly empty space with only a few specific activity values, providing little insight into the reaction in question. Why don't the authors place the lower half of the table in Supplementary Material and only show the first 4 rows.

Comment 2. On P. 13 the authors describe a catalytic mechanism, within the results section, but have very little data to back this up. It seems out of place in the manuscript. It seems to me that the authors should show the UV-Vis and structural data they have in the results section and then move figure 2 to the discussion to tie the paper together. They could refer to their data as well as results from related enzymes to strengthen the mechanistic argument.

Reviewer #3

(Remarks to the Author)

Thank you for addressing my comments in the response and revising the paper accordingly. I am satisfied with the revisions. Congratulations on the nice work.

Reviewer #4

(Remarks to the Author)

Overall, the work is thorough and makes significant contributions to understanding microbial haloalkane metabolism. However, I had a few points to note and requests for clarification:

1. The introduction would benefit from clarity regarding the heterologous expression system. As I read through the introduction section, it would help readers to know early on what heterologous system is being used and that these proteins were expressed in *E. coli*, rather than learning this later in the methods.
2. The introduction states that the Cdm system is "likely exclusive to anaerobic microorganisms" but doesn't briefly explain the rationale for this statement. A sentence or two explaining why this is expected would improve clarity.

3. I agree with the reviewer that parts of the work are somewhat speculative and lack a strong experimental foundation. Using a computational model to characterize CdmA function since the protein could not be produced is challenging. Does this also mean that you aren't able to achieve a full in vitro reconstitution of the pathway? This limitation should be acknowledged more explicitly.

4. The kinetic parameters for DCM show a K_m that is approximately 1000-fold higher than CM, and the authors interpret this as DCM being a non-physiological substrate. However, the V_{max} for DCM is actually slightly higher than CM. It would be helpful if the authors discussed what this means for the catalytic mechanism—is CdmB equally "efficient" at processing DCM once it binds, or does the high K_m primarily reflect something about substrate accessibility through the proposed channels?

5. In Table 3 (now Table 1 in the revised manuscript), several substrates show Co(III)-CdmC formation without CdmB being present (iodomethane, iodoethane, bromoethane). The authors explain this as "high reactivity with cobalt," but then also report specific activities for some of these substrates in the presence of CdmB. It's somewhat confusing how specific activity is being calculated when there's already spontaneous reaction occurring. While the footnote explains they subtract the background, I wonder if this is really measuring CdmB-catalyzed activity or just slightly accelerated spontaneous methylation. Additional clarification would be helpful.

Version 2:

Reviewer comments:

Reviewer #2

(Remarks to the Author)

The authors have address all of my comments. It should be published as is. Congratualtions on an excellent manuscript.

Reviewer #4

(Remarks to the Author)

The comments address all concerns and the work is good to be accepted.

Reviewer 2:

The manuscript entitled “Identification and characterisation of a long-elusive bacterial enzyme system for chloromethane dehalogenation” by Berhardt et al identified a gene cluster, cdmBCA, within *Acetobacterium dehalogenans* that encodes a novel corrinoid-dependent methyltransferase system, distinct from the characterized Cmu system used for chloromethane degradation in aerobic methylotrophs. The authors major findings are the identification of the cdmBCA gene cluster, the expression and purification of CdmB and CdmC and showed they could dehalogenate CM and DCM. They also solved the X-ray structure of BdmB. The work in this paper is thorough and thoughtfully presented; however, there are several shortcomings that likely prevent its publication in Nature Communications. I would suggest a more focused journal. My specific comments are below.

We thank the reviewer for the appreciation of our findings and the thorough presentation thereof. We would like to thank the reviewer for the useful suggestions, which helped us to improve the manuscript. The shortcomings mentioned by the reviewer were thoroughly addressed as explained below.

Comment 1. The authors report specific activities for CdmB towards CM and DCM as well as iodo, chloro, and bromo derivatives but fail to report k_{cat} or K_m values. These kinetic constants are required to understand substrate specificity and to gain insight into the mechanism of action. As it stands, Table 3 is mostly empty space with only a few specific activity values, providing little insight into the reaction in question.

We initially didn't determine k_{cat} and K_m as the activity assay requires large quantities of the corrinoid protein and the activity could not be measured at the organism's optimal temperature of 28°C (lines 562 ff): At 28°C, decreasing the CdmB concentration in an attempt to reduce the observed high reactivity resulted in gradual deactivation of the protein, as evidenced by incomplete methylation of the Co(I) state to the Co(III) state. However, we agree with the reviewer that the kinetic values would be insightful, as there are no other studies describing the catalytic properties, including substrate affinities, of a dehalogenating methyltransferase. We produced large quantities of the protein and measured activities for CdmB with chloromethane (natural substrate) and DCM with varying substrate concentrations (in technical triplicates) at the lower temperature of 20°C. We managed to obtain the kinetic parameters for these substrates and added this valuable information to the manuscript (lines 168 ff and 349 ff, Supplementary Fig. 6).

Comment 2. While interesting, the CdmB structure is poorly presented in Figure 3 making it difficult for the reading to glean any relevant information. While examining the iodo derivatives binding could be interesting, what the authors observe are multiple binding sites which provide little insight into how substrate binds for catalysis.

We obtained CdmB structures soaked with iodomethane and DCM. The structural data showed a distribution of both substrates at different occupancies in hydrophobic pockets. This behaviour is typical for the transport of gaseous substrates and provides a clear picture of how the protein scaffold CdmB will facilitate their diffusion inside the hydrophobic core. To our knowledge, this substrate

tunnelling via a hydrophobic network is novel for this type of enzyme. In support of our finding, we now also demonstrate that the tunnel is not restricted to CdmB from *A. dehalogenans*. For this, we generated and analysed AlphaFold 3 models from 16 CdmB homologues, and found a similar tunneling system (presented in the new Supplementary Fig. 10).

We also used AlphaFold 3 modelling to confirm that the CdmC Rossmann fold domain carrying the B₁₂ is indeed facing the CdmB hydrophobic core. However, due to the limitations of modelling, we could not provide a picture of how the Co(I)-loaded CdmC will interact with CdmB, as the current model resulted in multiple steric clashes (explained in line 261 f). This highlights the need for a conformational rearrangement of CdmC when it is locked in CdmB to properly position the corrinoid for catalysis, while preventing solvent molecules that might cross-react with the reactive Co(I). Therefore, we conclude that CdmB is contributing to the catalysis by (1) facilitating substrate diffusion, (2) offering a catalytic chamber for the nucleophilic attack, and (3) assisting CdmC in enhancing Co(I) reactivity (lines 378 ff and lines 279 ff below Fig. 4). This is supported by additional in vitro measurements showing that CM methylates Co(I)-free B₁₂ at similar rates in the presence and absence of CdmB, whereas CdmC blocks Co(I) reactivity with CM to prevent interference when CdmB is not bound to CdmC (now added in lines 158 ff).

To improve the information content for the readers, we split Figure 3 into two figures (now Figs. 3 and 4) and reorganised the colour coding to clarify and specify the provided information.

Fig. 3A is now an overview of the CdmB homodimer to describe its overall organisation.

Fig. 3B details the specific extension.

Fig. 3C is a simplified version of Fig. 3D and shows only hydrophobic tunnelling systems, introducing Fig. 3D and Fig. 3E-G dedicated to the substrate binding.

Figs. 4A and B present the AlphaFold 3 CdmBC model with the same colour coding as in Fig. 3. Finally, Fig. 4C show our proposed reaction mechanism (see comment 3).

Comment 3. On P. 22 the authors describe a catalytic mechanism, do not refer back to Figure 2, which makes it difficult to follow. What they describe does not appear to be supported by the data presented and is highly speculative. Additional data and a figure is needed to propose a reasonable mechanism.

The mechanism is indeed hypothetical. We mentioned this now more clearly in the manuscript (lines 378 ff). Our hypothesis is based on the knowledge of the reaction mechanism of corrinoid-dependent methyltransferases to date, alongside the findings obtained in this study (e.g. Fig. 3), which we now report in lines 279 ff and illustrate in Fig. 4C.

Comment 4. CdmA was also upregulated in response to CM (according to Figure 1). Is there a reason it was not also heterologously expressed?

We did not manage to produce soluble CdmA heterologously due to aggregation issues. However, methyl group transfer from CM to tetrahydrofolate (H₄F) was previously demonstrated in *Acetobacterium dehalogenans* in cell extract-based activity assays (Ref 21), before the respective enzyme system was identified (Line 135). Furthermore, we now added Supplementary Fig. 5 showing a

structural overlay of CdmA AlphaFold 3 model from *A. dehalogenans* with the methyltransferase involved in O-demethylation from *Desulfitobacterium hafniense* CdmA with bound H4F. Most of the residues of CdmA involved in H4F binding are very well conserved. ***Therefore, we think obtaining CdmA and demonstrating H₄F methylation is not essential for this study.***

Minor comments

Within the introduction the authors fail to mention of the toxicity of CM and DCM to humans and other animals. While their focus is on ozone depletion and climate change (very important issues), acute toxicities of these compounds would further motivate and empathize their research. It may also be valuable to mention current methods for removing CM and DCM from the environment.

Thanks for the suggestion. We added the following sentence to the Introduction lines 43 ff.: “CM and DCM can be highly toxic and pose significant health risks to humans and animals, targeting the central nervous system, liver, and kidneys (Tsai et al 2017 (10.3390/toxics5040023), Kumari et al. 2024 (<https://doi.org/10.1007/s44339-024-00012-8>). Various methods can be employed to remove these contaminants from the environment, particularly from water and soil, including physical (air stripping, adsorption), biological (microbial degradation), and chemical (oxidation, electrocatalysis) techniques, which are often used in combination. However, further research is necessary to optimize and improve the efficiency of these removal strategies, as well as to develop new and more effective methods for mitigating the environmental impact of these pollutants (Shestakova et al 2013 (<https://doi.org/10.1016/j.chemosphere.2013.07.022>), Lemus et al, 2012 (10.3390/toxics5040023).”

Line 191: Unnecessary close parenthesis present

We corrected this.

Line 201: Is there a reason different concentrations for each potential substrate were used? How was each concentration decided on as the Km values are unknown?

The used haloalkanes are toxic volatile compounds and their handling is not trivial. We used very small volumes of the pure liquid compounds (1-5 µl) to obtain final concentrations of at least 2 mM for all of them, in accordance with the concentrations used in other methyltransferase studies, and in order to ensure saturating substrate concentrations.

Line 242: “Diluted with one volume.” What does “one volume” mean in this context?

Here we mean column volume. It has been specified now in the text (line 626).

Line 369: Is there a proposed reason large compounds are not substrates (size of active site)?

The active site cavity is rather spacious. It is very likely the size of the substrate channels that determines the substrate spectrum for this enzyme especially in comparison to closely related O-demethylases, which have a way broader substrate entry channel/cavity. As mentioned in the answer

to Comment 2, CdmB has specific hydrophobic channels of defined diameters, to act as a selective filter recognising only small molecules and avoiding interference (Fig. 3; Supplementary Fig. 10).

Line 380-1: This hypothesis potentially clashes with the data presented in Table 3. If DCM is inhibiting the Cdm system, shouldn't specific activity be significantly lower for DCM than CM? If DCM is irreversibly binding CdmC, that would effectively decrease the concentration of CdmC in the reaction.

In the activity assay we use the corrinoid protein (here CdmC) in excess as done in most methyltransferase studies to follow methylation of the cobalt ion of the corrinoid cofactor. The transfer of the methyl or alkyl group from the haloalkane to the cobalt ion of CdmC can be followed spectroscopically with help of the cobalt redox chemistry and subsequent color change. That means at the end of the activity assay the methyl or alkyl group is bound to the cobalt ion and we are not testing in this assay if the methyl or alkyl group is subsequently removed from the cobalt (which in case of CM would require CdmA and H4F addition). Therefore, our hypothesis does not clash with the obtained results.

Line 382: Were both those that are substrates (bromopropane/bromobutane) and those which are not substrates (bromopentane) inhibitory or just one of these groups?

All of these compounds were inhibitory. We show this in Supplementary Fig. 7 (SI Fig. 5 in the original manuscript).

Reviewer 3:

This paper by Berhardt et al. describes the identification and heterologous production of the proteins/enzymes involved in anaerobic dehalogenation/demethylation of chloromethane by *Acetobacterium dehalogenens* along with demonstration of the activity, substrate specificity, crystal structure, and substrate binding of the enzyme complex. I find this study to be a highly noteworthy contribution to the fields of dehalogenation research, anaerobic enzymology, and structural biology. The work is well-planned and executed and the work appears to be well-described for reproducibility purposes. It is a much needed and anticipated discovery that will prove to be a significant contribution to science.

We would like to thank the reviewer for appreciating our study and evaluating it as highly noteworthy contribution to science, well executed and well-described. We are grateful for the following helpful suggestions.

I had only minor concerns/questions about the study:

1. The hypothesis or model that CdmB acts merely to channel and position CM for nucleophilic attack by Co(I)-CdmC is fascinating. This begs the question of whether CdmB and CdmC can function independently. The authors indicate in the results "Notably, iodomethane, iodoethane, and bromoethane reacted with CdmC even in the absence of CdmB, likely due to their high reactivity with cobalt." It was not clear to me whether Co(I)-CdmC could spontaneously react with CM in the absence of CdmB (I apologize if I missed that in the text). If CdmB merely guides and positions the CM for nucleophilic Co(I) attack, one would assume that there would be some low level of activity of CdmC methylation in the absence of CdmB. The spectra they showed seem to indicate that the Co(I) state of CdmC is relatively stable so perhaps the hydrophobic pocket created by CdmB to protect the Co(I) is not entirely needed.

We did not observe a reaction of CdmC with CM or DCM in the absence of CdmB under in vitro conditions (see Table 1, Table 3 in the original manuscript), as these compounds are less reactive than iodomethane, iodoethane and bromoethane. However, for free cobalamin, we observed a reaction of CM with cobalamin without CdmB addition (see answer below). The finding that CM reacts with free cobalamin but not with CdmC in the absence of CdmB may indicate that only through the conformational changes of CdmC after binding to CdmB, CM can react with the cobalt ion of the corrinoid cofactor. This indicates that the role of CdmB is not solely to provide a hydrophobic pocket to protect Co(I) (see lines 158 ff and Fig. 4, and our answer to comment 2 of reviewer 2 above). With more reactive haloalkanes such as iodoethane the substrate reacted with CdmC in absence of CdmB. However, only CM seems to be a physiological substrate of the Cdm system, as explained in the text. Furthermore, we also show now that the hydrophobic tunneling systems seem to be conserved in close homologues (new Supplementary Fig. 10).

2. In addition, I don't believe the authors tested the ability of CdmB to use CM to methylate Co(I)-free B₁₂, which some MT1 enzymes can do with their specific substrates. Perhaps the authors have tried these experiments already and did not report them. If not, they seem to be quick and easy experiments to try that would help further elucidate the mechanism of this enzyme system and potentially support the hypothesis they proposed.

Thanks for the valuable feedback. We performed the suggested assay and found that the addition of CM leads to methylation of Co(I)-free B₁₂ in vitro, both in the presence and absence of CdmB, with similar rates. It is an interesting finding that CM methylates free Coenzyme B₁₂ but not CdmC, indicating that the protein backbone of CdmC prevents direct reaction with chloromethane. We have added this experiment to the Methods section (lines 600 ff) and described the results in lines 158 ff. Based on this experiment, we further concluded that the interaction between CdmB and CdmC might rearrange the CdmC Rossmann domain and the B₁₂ in a way to enhance Co(I) reactivity. Therefore, when CdmC is locked on CdmB, the haloalkanes diffusing through the hydrophobic network will directly face the Co(I) in an adequate catalytic chamber to perform the nucleophilic attack. This is now illustrated in Fig. 4C.

We would like to thank all reviewers for evaluating our revised manuscript and for providing helpful and constructive suggestions. We carefully revised the manuscript to address all reviewers' comments as detailed below (in bold font).

REVIEWER COMMENTS

Reviewer #2 (Remarks to the Author):

The authors address most of my comments so I suggest it be published in Nature Communications with some minor revisions.

We thank the reviewer for valuable feedback and comments. We fully addressed the additional points that were raised.

Comment 1. As it stands, Table 3 is still mostly empty space with only a few specific activity values, providing little insight into the reaction in question. Why don't the authors place the lower half of the table in Supplementary Material and only show the first 4 rows.

We thank the reviewer for this insightful suggestion. We deleted the lower half of Table 1 (all rows for non-haloalkanes). We now refer to corresponding data in the text as follows: 'A specificity of the Cdm system for haloalkanes was clearly evident from the fact that MACs, methylated/methoxylated alcohols, dimethyl disulphide, and methylamines were not transformed ((Source data: table 1)).' (Lines 188-190).

Comment 2. On P. 13 the authors describe a catalytic mechanism, within the results section, but have very little data to back this up. It seems out of place in the manuscript. It seems to me that the authors should show the UV-Vis and structural data they have in the results section and then move figure 2 to the discussion to tie the paper together. They could refer to their data as well as results from related enzymes to strengthen the mechanistic argument.

We thank the reviewer for these helpful suggestions. We carefully revised this part of the results section to make clear that the proposed reaction sequence in Fig. 2A is based on our UV-vis spectroscopy results (shown in Fig. 2B): 'UV-vis spectroscopy using CdmBC proteins and AE from A. dehalogenans revealed that AE-mediated reduction of Co(II)-CdmC to Co(I)-CdmC is followed by a MTI CdmB-mediated reaction with CM to form CH₃-Co(III)-CdmC' (lines 134 ff). Fig 2A is important to understand the following results section and the discussion. We agree with the reviewer that Fig. 2C is not necessary in the results section and moved it to Supplementary information (now Supplementary Figure 13). Corresponding text was moved to the discussion (lines 432-439).

Reviewer #3 (Remarks to the Author):

Thank you for addressing my comments in the response and revising the paper accordingly. I am satisfied with the revisions. Congratulations on the nice work.

We are pleased that the reviewer was satisfied with the revised manuscript, thank you very much for your appreciation of our work.

Reviewer #4 (Remarks to the Author):

Overall, the work is thorough and makes significant contributions to understanding microbial haloalkane metabolism. However, I had a few points to note and requests for clarification:

We would like to thank the reviewer for appreciation of our work and for insightful comments and suggestions. We addressed and comprehensively clarified the few points that were raised.

1. The introduction would benefit from clarity regarding the heterologous expression system. As I read through the introduction section, it would help readers to know early on what heterologous system is being used and that these proteins were expressed in *E. coli*, rather than learning this later in the methods.

We now explicitly added the information on the proteins that were produced heterologously in *Escherichia coli* (line 65).

2. The introduction states that the Cdm system is "likely exclusive to anaerobic microorganisms" but doesn't briefly explain the rationale for this statement. A sentence or two explaining why this is expected would improve clarity.

We agree with the reviewer that our wording was not sufficiently clear. We have now clarified the statement as follows: 'Our results suggest that this system is likely exclusive to anaerobic microorganisms, since cdmB-like genes are encoded by anaerobic members of the Bacillota phylum and of the Asgardarchaeota phylum considered to comprise mainly anaerobic Archaea.' (Lines 70-71)

3. I agree with the reviewer that parts of the work are somewhat speculative and lack a strong experimental foundation. Using a computational model to characterize CdmA function since the protein could not be produced is challenging. Does this also mean that you aren't able to achieve a full in vitro reconstitution of the pathway? This limitation should be acknowledged more explicitly.

The reviewer is correct that we could not demonstrate H₄F methylation via CdmA. We now clarified that we could not produce CdmA in *E. coli* and that the CdmA-mediated reaction of H₄F was not directly

demonstrated experimentally in our study (line 139-140). However, we would like to point out that transfer of the methyl group of chloromethane to tetrahydrofolate (H₄F) was previously reported in A. dehalogenans using cell extract-based activity assays (as mentioned in lines 141 ff). Also, use of H₄F as the major C1-carrier in central metabolism is well-known for a broad range of bacterial methyltransferase systems (e.g., refs 28-31). Therefore, transfer of the methyl group to H₄F has widely acknowledged experimental precedent.

Moreover, the AlphaFold 3 model of CdmA with bound H₄F/CH₃-H₄F shown in Supplementary Figure 5 further supports this notion, as the modelled binding site of CdmA conserves all molecular determinants for H₄F specificity. We now mention this more clearly and explicitly in the text (lines 143-145).

4. The parameters for DCM show a K_m that is approximately 1000-fold higher than CM, and the authors interpret this as DCM being a non-physiological substrate. However, the V_{max} for DCM is actually slightly higher than CM. It would be helpful if the authors discussed what this means for the catalytic mechanism—is CdmB equally "efficient" at processing DCM once it binds, or does the high K_m primarily reflect something about substrate accessibility through the proposed channels?

We thank the reviewer for raising these questions and we agree that these points required clarification. We recently confirmed and consolidated our data and their analysis by experimentally measuring by gas chromatography the CM concentrations used in our determination of kinetic parameters of Cdm. The consolidated K_m values for CM (lines 204 and 385 ff) show that the K_m value for CM is 24-fold lower than that for DCM. Thus, and as pointed out by the reviewer on the basis of the very similar V_{max} of CdmB for CM and DCM, activity of CdmB with DCM is likely as efficient as with CM under saturating conditions of substrate. We carefully rewrote this passage accordingly (lines 206-209). In addition, we discussed the difference in affinity between CM and DCM and the possible role of CdmB in enhancing specificity for CM in the discussion (lines 397-398, 411-412), as also suggested by the reviewer.

5. In Table 3 (now Table 1 in the revised manuscript), several substrates show Co(III)-CdmC formation without CdmB being present (iodomethane, iodoethane, bromoethane). The authors explain this as "high reactivity with cobalt," but then also report specific activities for some of these substrates in the presence of CdmB. It's somewhat confusing how specific activity is being calculated when there's already spontaneous reaction occurring. While the footnote explains they subtract the background, I wonder if this is really measuring CdmB-catalyzed activity or just slightly accelerated spontaneous methylation. Additional clarification would be helpful.

We thank the reviewer very much for pointing out these issues. We agree that Table 1 and associated text needed to better emphasise that conversion of CM requires CdmB. Table 1 and associated text were therefore carefully and thoroughly revised to clarify this point, the nature of the reactions of different haloalkanes with the Cdm system, and also of non-enzymatic reactions of CM and other haloalkanes with cobalamin (lines 174-187).

REVIEWER COMMENTS

Reviewer #2:

The authors have address all of my comments. It should be published as is. Congratualtions on an excellent manuscript.

We would like to thank Reviewer 2 for their satisfaction with our revised manuscript.

Reviewer #4:

The comments address all concerns and the work is good to be accepted.

We would like to thank Reviewer 4 for confirming that all concerns have been addressed by our comments.